# FAST, GRADIENT-FREE LEARNING
# WITH CONDITIONAL MIXTURE NETWORKS

## ABSTRACT

Bayesian methods are known to address some limitations of standard deep learning, such as the lack of calibrated predictions and uncertainty quantification. However, they can be computationally expensive as model and data complexity increase. Fast variational methods can reduce the computational requirements of Bayesian methods by eliminating the need for gradient descent or sampling, but are often limited to simple models. We demonstrate that conditional mixture networks (CMNs), a probabilistic variant of the mixture-of-experts (MoE) model, are suitable for fast, gradient-free inference and can solve complex classification tasks, thus balancing the expressiveness and scalability of neural networks with the probabilistic benefits of Bayesian methods . By exploiting conditional conjugacy and Pólya-Gamma augmentation, we furnish Gaussian likelihoods for the weights of both the experts and the gating network. This enables efficient variational updates using coordinate ascent variational inference (CAVI), avoiding traditional gradient-based optimization. We validate this approach by training two-layer CMNs on standard benchmarks from the UCI repository. Our method, CAVI-CMN, achieves competitive and often superior predictive accuracy compared to maximum likelihood estimation (MLE) with backpropagation, while maintaining competitive runtime and full posterior distributions over all model parameters. Moreover, as input size or the number of experts increases, computation time scales competitively with MLE and other gradient-based solutions like black-box variational inference (BBVI), making CAVI-CMN a promising tool for deep, fast, and gradient-free Bayesian networks.

## 1 INTRODUCTION

Modern machine learning methods attempt to learn functions of complex data (e.g., images, audio, text) to predict information associated with that data, such as discrete labels in the case of classification (Bernardo et al., 2007). Deep neural networks (DNNs) have demonstrated success in this domain, owing to their universal function approximation properties (Park & Sandberg, 1991) and the soft regularization inherited from stochastic gradient descent learning via backpropagation (Amari, 1993). However, despite its computational efficiency, accuracy, and scalability to increasingly large datasets and models, DNNs trained this way do not provide well-calibrated predictions or uncertainty estimates, and practitioners typically utilize post-hoc calibration methods on validation datasets (Wang et al., 2021; Shao et al., 2020). This limits the applicability and reliability of using DNNs in safety-critical applications like autonomous driving, medicine, and disaster response (Papamarkou et al., 2024), where uncertainty-sensitive decision-making is required.

Bayesian machine learning addresses the issues of poor calibration and uncertainty quantification by offering a probabilistic framework that casts learning model parameters $\boldsymbol{\theta}$ as a process of inference - namely, calculating a posterior distribution over model parameters $p\left(\boldsymbol{\theta} \mid \mathcal{D}\right)$, given observed data $\mathcal{D}$ (e.g., a data set of input-output pairs $\mathcal{D} = \left((\boldsymbol{x}_1, \boldsymbol{y}_1), \ldots, (\boldsymbol{x}_n, \boldsymbol{y}_n)\right)$). The resulting posterior distribution captures both expectations about model parameters $\boldsymbol{\theta}$ and their uncertainty. The uncertainty is then incorporated in predictions $p(\boldsymbol{y})$ that are, in principle, well-calibrated to new datapoints coming from the same set. This probabilistic treatment allows methods like Bayesian neural networks (BNNs) (Hernández-Lobato & Adams, 2015) to maintain the expressiveness of deep neural networks while also encoding uncertainty over network weights and thus the network's predictions. However,

these methods are often accompanied by an increase in computational cost and still involve running gradient descent or generating samples (Izmailov et al., 2021).

In this paper we introduce a gradient-free variational learning algorithm for a probabilistic variant of a two-layer, feedforward neural network — the conditional mixture network or CMN — and measure its performance on supervised learning benchmarks. This method rests on coordinate ascent variational inference (CAVI) (Wainwright et al., 2008; Hoffman et al., 2013) and hence we name it CAVI-CMN. We compare CAVI-CMN to maximum likelihood estimation and two other Bayesian estimation techniques: the No U-Turn Sampler (NUTS) variant of Hamiltonian Monte Carlo (Hoffman et al., 2014) and black-box variational inference (Ranganath et al., 2014). We demonstrate that CAVI-CMN maintains the predictive accuracy and scalability of an architecture-matched feedforward neural network fit with maximum likelihood estimation (*i.e.*, gradient descent via backprop-agation), while maintaining full distributions over network parameters and generating calibrated predictions, as measured in relationship to state-of-the-art Bayesian methods like NUTS and BBVI. Unlike other Bayesian (NUTS, BBVI) and non-Bayesian gradient based approaches (MLE), CAVI-CMN achieves SOTA-like performance with absolute runtime comparable to a backpropagation-based MLE approach, thanks to the fast convergence properties of coordinate ascent variational inference.

We summarize the contributions of this work below:

- We introduce and derive a variational inference scheme for the conditional mixture network, which we term CAVI-CMN. This relies on the use of conjugate priors for the linear experts and Pólya-Gamma augmentation (Polson et al., 2013) for the gating network and the final softmax layer.

- CAVI-CMN matches, and sometimes exceeds, the performance of maximum likelihood estimation (MLE) in terms of predictive accuracy, while maintaining the benefits of a full Bayesian approach, yielding well calibrated models that quantify uncertainty. This is shown across a suite of 8 different supervised classification tasks (2 synthetic, 6 real).

- CAVI-CMN displays all the benefits explained above while requiring drastically less time to converge and overall runtime than the other state-of-the-art Bayesian methods like NUTS and BBVI.

The rest of this paper is organized as follows: first, we discuss related works include the MoE architecture and existing (Bayesian and non-Bayesian approaches) to fitting these models. We then introduce the probabilistic conditional mixture model and derive a variational inference algorithm for optimizing posterior distributions over its latent variables and parameters. We present experimental results comparing the performance of CAVI-based conditional mixture models with sampling based methods, such as BBVI, NUTS, and traditional MLE based estimation, where gradients of the log likelihood are computed using backpropagation and used to update the network's parameters. Finally, we discuss the implications of these findings and potential directions for future research.

## 2 RELATED WORK

The Mixture-of-Experts (MoE) architecture is a close relative of the CMN model we introduce here. Jacobs et al. (1991) originally introduced MoEs as a way to improve the performance of neural networks by combining the strengths of multiple specialized models (Gormley & Frühwirth-Schnatter, 2019). MoE models process inputs by averaging the predictions of individual learners or experts, where each expert's output is weighted by a different mixing coefficient before the averaging. The fundamental idea behind MoE is that the input space can be partitioned in such a way that different experts (models) can be trained to excel in different regions of this space, with a gating network determining the appropriate expert (or combination of experts) for each input. This leads to composable (and sometimes interpretable) latent descriptions of arbitrary input-output relationships (Eigen et al., 2013), further bolstered by the MoE's capacity for universal function approximation (Nguyen et al., 2016; Nguyen & Chamroukhi, 2018). Indeed, the powerful self-attention mechanism employed by transformers has has demonstrated the power and flexibility of MoE models (Movellan & Gabbur, 2020). Non-Bayesian approaches to MoE typically rely on maximum likelihood estimation (MLE) (Jacobs et al., 1991; Jordan & Jacobs, 1994), which can suffer from overfitting and poor

generalization due to the lack of regularization mechanisms (Bishop & Svens`kn, 2003), especially in low data size regimes.

To address these issues, Bayesian approaches to MoE have been developed, which incorporate prior information and yield posterior distributions over model parameters (Bishop & Svens`kn, 2003; Mossavat & Amft, 2011). This Bayesian treatment enables the estimation of model evidence (log marginal-likelihood) and provides a natural framework for model comparison and selection (Svensén, 2003; Zens, 2019). Bayesian MoE models offer significant advantages, such as improved robustness against overfitting and a better understanding of uncertainty in predictions. However, they also introduce computational challenges, particularly when dealing with high-dimensional data and complex model structures.

The introduction of the Pólya-Gamma (PG) augmentation technique in Polson et al. (2013) enabled a range of novel and more computationally efficient algorithms for Bayesian treatment of MoE models (Linderman et al., 2015; He et al., 2019; Sharma et al., 2019; Viroli & McLachlan, 2019; Zens et al., 2023). Here we complement these past works, which mostly rest on improving sampling methods with PG augmentation, by introducing a closed-form update rules for MoE's with linear experts in the form of coordinate ascent variational inference (CAVI).

## 3 METHODS

In this section we first motivate the use of conditional mixture models for supervised learning, and then introduce the conditional mixture network (CMN), the probabilistic model whose properties and capabilities we demonstrate in the remainder of the paper.

### 3.1 CONDITIONAL MIXTURES FOR FUNCTION APPROXIMATION

Feedforward neural networks are highly expressive, approximating nonlinear functions through sequences of nonlinear transformations, but the posterior distributions over their weights are intractable, requiring expensive techniques like MCMC or variational inference (MacKay, 1992; Blundell et al., 2015; Daxberger et al., 2021).

We circumvent these problems by focusing on the Mixture-of-Experts (MoE) models (Jacobs et al., 1991), and particularly a variant of MoE that is amenable to gradient free CAVI parameter updates. MoEs can be made tractable to gradient-free CAVI when the expert likelihoods are constrained to be members of the exponential family (see Section 2 for more details on the MoE architecture), and when the gating network is formulated in such a way to allow exact Bayesian inference (through lower bounds on the log-sigmoid likelihood (Jaakkola & Jordan, 1997; Bishop & Svens`kn, 2003) or Pólya-Gamma augmentation (Polson et al., 2013)).

The MoE can be reformulated probabilistically as a mixture model by introducing a latent assignment variable, $z^n$, leading to a joint probability distribution of the form

$$p(Y, Z, \Theta) = p(\theta_{1:K})p(\pi) \prod_{i=1}^{N} p(y^n | z^n, \theta_{1:K})p(z^n | \pi) \, ,$$

where $y^n$ is an observation, $\Theta = \{\theta_{1:K}, \pi\}$, $p_k(y^n | \theta_k)$ is the $k^{\text{th}}$-component's likelihood and $z^n$ is a discrete latent variable that assigns the $n^{\text{th}}$ datapoint to one of the $K$ mixture components, i.e. $p_k(y^n | \theta_k) = p(y^n | z^n = k, \theta_{1:K})$. For instance, if each 'expert' likelihood $p_k(y^n | \theta_k)$ is a Gaussian distribution, then the MoE becomes a Gaussian Mixture Model, where $\theta_k = (\mu_k, \Sigma_k)$.

The problem of learning the model's parameters, then becomes one of doing inference over the latent variables $Z$ and parameters $\Theta$ of the mixture model. However, mixture models are generally not tractable for exact Bayesian inference, so some form of approximation or sampling-based scheme is required to obtain full posteriors over their parameters. However, if each expert (i.e., likelihood distribution) in the MoE belongs to the exponential family, the model becomes *conditionally conjugate*. This allows for derivation of exact fixed-point updates to an approximate posterior over each expert's parameters. The approach we propose, CAVI-CMN, does exactly this – we take advantage of the conditional conjugacy of mixture models, along with an augmentation trick for the the gating network, to make all parameters amenable to an approximate Bayesian treatment. The

conditionally-conjugate form of the model allows us to use coordinate ascent variational inference to obtain posteriors over the weights of both the individual linear experts and the gating network (Wainwright et al., 2008; Hoffman et al., 2013; Blei et al., 2017), without resorting to costly gradient or sampling computations.

Going forward we use the term *conditional mixture networks* (CMN) to emphasize (1) the discriminative nature of proposed application of this approach, where the model is designed to predict an output $y$ given an input $x$ and (2) the fact that individual MoE layers can be stacked hierarchically into a feedforward architecture. This makes CMNs particularly suitable for tasks such as supervised classification and regression, where the goal is effectively that of function approximation; predict some output variable $y$ given input regressors $x$.

### 3.2 CONDITIONAL MIXTURE NETWORK OVERVIEW

The conditional mixture network maps from a continuous input vector $\boldsymbol{x}_0 \in \mathbb{R}^d$ to its label $y \in \{1, \ldots, L\}$. This is achieved with two layers: a conditional mixture of linear experts, which outputs a joint continuous-discrete latent $\left(\boldsymbol{x}_1 \in \mathbb{R}^h, z_1 \in \{1, \ldots, K\}\right)$ and a multinomial logistic regression, which maps from the continuous latent $\boldsymbol{x}_1$ to the corresponding label $y$. The probabilistic mapping can be described in terms of the following operations:

$$z_1 \sim \text{Mult}\left(z_1; \boldsymbol{x}_0, \boldsymbol{\beta}_0\right)$$
$$\boldsymbol{x}_1 = \boldsymbol{A}_{z_1} \cdot [\boldsymbol{x}_0; 1] + \boldsymbol{u}_{z_1}, \qquad \boldsymbol{u}_{z_1} \sim N(\boldsymbol{0}, \boldsymbol{\Sigma}_{z_1})$$
$$y \sim \text{Mult}\left(y; \boldsymbol{x}_1, \boldsymbol{\beta}_1\right)$$

where we pad the input variable $\boldsymbol{x}_0$ with a constant value set to 1, to absorb the bias term within the mapping matrix $\boldsymbol{A}_{z_1} \in \mathbb{R}^{h \times d+1}$, and where $\text{Mult}\left(z; \boldsymbol{x}, \boldsymbol{\beta}\right)$ denotes a multinomial distribution parameterized with a regressor $\boldsymbol{x}$ and logistic regression coefficients $\boldsymbol{\beta}$. Note that for every pair of regressors and labels $(\boldsymbol{x}_0^n, y^n)$, we assume a corresponding pair of latent variables $(\boldsymbol{x}_1^n, z_1^n)$. Written in this way, it becomes clear than CMN is a mixture of linear transforms that is capable of modeling non-linear transfer functions via a piecewise linear approximation.

In order to obtain a normally distributed posterior over the multinomial logistic regression weights, $\boldsymbol{\beta}_0$ and $\boldsymbol{\beta}_1$, we use Pólya-Gamma augmentation (Polson et al., 2013; Linderman et al., 2015) applied to the stick breaking construction for the multinomial distribution:

$$p\left(z = k | \boldsymbol{\beta}, \boldsymbol{x}\right) = \pi_k\left(\boldsymbol{\beta}, \boldsymbol{x}\right) \prod_{j=1}^{k-1} \left(1 - \pi_j\left(\boldsymbol{\beta}, \boldsymbol{x}\right)\right)$$
$$\pi_j\left(\boldsymbol{\beta}, \boldsymbol{x}\right) = \frac{1}{1 + \exp\left\{-\boldsymbol{\beta}_j \cdot [\boldsymbol{x}; 1]\right\}}, \forall j < K \tag{1}$$
$$\pi_K = 1$$

where for the gating network (input layer) we will have coefficients of dimension $\boldsymbol{\beta}_0 \in \mathbb{R}^{K-1 \times d}$, and for the output likelihood coefficients of dimension $\boldsymbol{\beta}_1 \in \mathbb{R}^{L-1 \times h}$.

### 3.3 GENERATIVE MODEL FOR THE CONDITIONAL MIXTURE NETWORK

Given a set of labels $Y = \left\{y^1, y^2, ..., y^N\right\}$, and regressors $\boldsymbol{X}_0 = \left\{\boldsymbol{x}_0^1, \boldsymbol{x}_0^2, ..., \boldsymbol{x}_0^N\right\}$, that define i.i.d input-output pairs $\boldsymbol{x}_0^n, y^n$, we write the joint distribution over labels $Y$, latents $\boldsymbol{X}_1, Z_1$, and parameters $\boldsymbol{\Theta}$ as:

$$p(\boldsymbol{Y}, \boldsymbol{X}_1, Z_1, \boldsymbol{\Theta} | \boldsymbol{X}_0) = p(\boldsymbol{\Theta}) \prod_{n=1}^N p_{\boldsymbol{\beta}_1}\left(y^n | \boldsymbol{x}_1\right) p_{\boldsymbol{\lambda}_1}\left(\boldsymbol{x}_1^n | \boldsymbol{x}_0^n, z_1^n\right) p_{\boldsymbol{\beta}_0}\left(z_1^n | \boldsymbol{x}_0^n\right)$$
$$p(\boldsymbol{\Theta}) = p(\boldsymbol{\beta}_1) p(\boldsymbol{\beta}_0) p(\boldsymbol{\lambda}_1) \tag{2}$$
$$= \prod_{l=1}^{L-1} p(\boldsymbol{\beta}_{l,1}) \prod_{k=1}^{K-1} p(\boldsymbol{\beta}_{k,0}) \prod_{j=1}^K p\left(\boldsymbol{A}_j, \boldsymbol{\Sigma}_j^{-1}\right)$$

Note that this model structure, with input and target variables, is often referred to as a *discriminative* model, as opposed to a *generative* model (Bernardo et al., 2007). However, we use the term

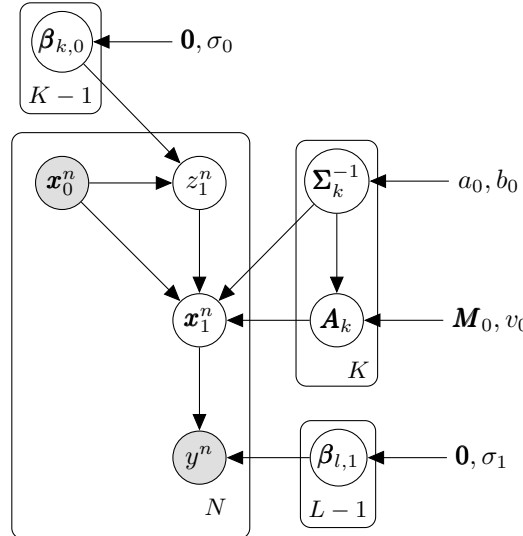

Figure 1: A Bayesian network representation of the two-layer conditional mixture network, with input-output pairs $\boldsymbol{x}_0^n, y^n$ and latent variables $\boldsymbol{x}_1^n, z_1^n$. Observations are shaded nodes, while latents and parameters are transparent. Prior hyperparameters are shown without boundaries.

generative model to emphasize the fact that the model contains priors over latent variables $(\boldsymbol{X}_1, Z_1)$, and parameters $\left(\boldsymbol{\Theta} = \left(\boldsymbol{\beta}_{1:L-1,1}, \boldsymbol{\beta}_{1:K-1,0}, \boldsymbol{A}_{1:K}, \boldsymbol{\Sigma}_{1:K}^{-1}\right)\right)$, and that we are estimating posteriors over these quantities, by maximizing a lower bound on marginal likelihood of the observed target variables $Y$. Note that going forward, we will sometimes use $\boldsymbol{\lambda}_1$ as notational shorthand for the parameters $\boldsymbol{A}_{1:K}, \boldsymbol{\Sigma}_{1:K}^{-1}$ of the first layer's linear experts.

We specify the following conditionally conjugate priors for the parameters of the two-layer CMN:

$$p\left(\boldsymbol{A}_k | \boldsymbol{\Sigma}_k^{-1}\right) = \mathcal{MN}\left(\boldsymbol{A}_k; \boldsymbol{M}_0, \boldsymbol{\Sigma}_k, v_0 \boldsymbol{I}_{d+1}\right)$$

$$p\left(\boldsymbol{\Sigma}_k^{-1} \equiv \operatorname{diag}\left(\boldsymbol{\sigma}_k^{-2}\right)\right) = \prod_{i=1}^{h} \Gamma\left(\sigma_{k,i}^{-2}; a_0, b_0\right)$$

$$p\left(\boldsymbol{\beta}_{k,0}\right) = \mathcal{N}\left(\boldsymbol{\beta}_{k,0}; \boldsymbol{0}, \sigma_0^2 \boldsymbol{I}_{d+1}\right)$$

$$p\left(\boldsymbol{\beta}_{l,1}\right) = \mathcal{N}\left(\boldsymbol{\beta}_{l,1}; \boldsymbol{0}, \sigma_1^2 \boldsymbol{I}_{h+1}\right)$$

(3)

where we fixed the prior mean matrix of the linear transformation $\boldsymbol{A}_k$ to be a matrix of zeros: $\boldsymbol{M}_0 = \boldsymbol{0}$. Other hyperparameters of the priors are described in Appendix C.1. In the following section we introduce a mean-field variational inference scheme we use for performing inference and learning in the two-layer CMN.

### 3.4 COORDINATE ASCENT VARIATIONAL INFERENCE WITH CONJUGATE PRIORS

In this section we describe a variational approach for inverting the probabilistic model described in Equation (2) and computing an approximate posterior over latents and parameters specified as

$$p\left(\boldsymbol{X}_1, Z_1, \boldsymbol{\Theta} | Y, \boldsymbol{X}\right) = \frac{p\left(Y, \boldsymbol{X}_1, Z_1, \boldsymbol{\Theta}, \boldsymbol{X}\right)}{p\left(Y | \boldsymbol{X}\right)} \approx q\left(\boldsymbol{\Theta}\right) \prod_{n=1}^{N} q\left(z_1^n\right) q\left(\boldsymbol{x}_1^n | z_1^n\right) \qquad (4)$$

where $q\left(\boldsymbol{x}_1^n | z_1^n\right)$ corresponds to a component specific multivariate normal distribution, and $q\left(z_1^n\right)$ to a multinomial distribution. Importantly, the approximate posterior over parameters $q\left(\boldsymbol{\Theta}\right)$ further

factorizes (Svensén, 2003) as

$$q\left(\mathbf{\Theta}\right) = \prod_{l=1}^{L-1} q\left(\boldsymbol{\beta}_{l,1}\right) \prod_{k=1}^{K-1} q\left(\boldsymbol{\beta}_{k,0}\right) \underbrace{\prod_{j=1}^{K} q\left(\boldsymbol{A}_j, \Sigma_j^{-1}\right)}_{=q(\boldsymbol{\lambda}_1)}$$

$$q\left(\boldsymbol{\beta}_{l,1}\right) = \mathcal{N}\left(\boldsymbol{\beta}_{l,1}; \boldsymbol{\mu}_{l,1}, \boldsymbol{\Sigma}_{l,1}\right)$$

$$q\left(\boldsymbol{\beta}_{l,0}\right) = \mathcal{N}\left(\boldsymbol{\beta}_{l,0}; \boldsymbol{\mu}_{k,0}, \boldsymbol{\Sigma}_{k,0}\right) \tag{5}$$

$$q\left(\boldsymbol{A}_j | \Sigma_j^{-1}\right) = \mathcal{MN}\left(\boldsymbol{A}_j; \boldsymbol{M}_j, \boldsymbol{\Sigma}_j, \boldsymbol{V}_j\right)$$

$$q\left(\boldsymbol{\Sigma}_j^{-1}\right) = \prod_{i=1}^{h} \Gamma\left(\sigma_{i,j}^{-2}; a_j, b_{i,j}\right)$$

The above form of the approximate posterior allows us to define tractable conditionally conjugate updates for each factor. This becomes evident from the following expression for the evidence lower-bound (ELBO) on the marginal log likelihood

$$\mathcal{L}(q) = \mathbb{E}_{q(\boldsymbol{X}_1, \boldsymbol{Z}_1) q(\boldsymbol{\Theta})} \left[ \sum_{n=1}^{N} \ln \frac{p_{\boldsymbol{\Theta}}\left(y^n, \boldsymbol{x}_1^n, z_1^n | \boldsymbol{x}_0^n\right)}{q\left(z_1^n\right) q\left(\boldsymbol{x}_1^n | z_1^n\right)} \right] + \mathbb{E}_{q(\boldsymbol{\Theta})} \left[ \ln \frac{p\left(\boldsymbol{\beta}_1\right) p\left(\boldsymbol{\beta}_0\right) p\left(\boldsymbol{\lambda}_1\right)}{q\left(\boldsymbol{\beta}_1\right) q\left(\boldsymbol{\beta}_0\right) q\left(\boldsymbol{\lambda}_1\right)} \right] \tag{6}$$

We maximize the ELBO using an iterative update scheme for the parameters of the approximate posterior, often referred to as variational Bayesian expectation maximization (VBEM) (Beal, 2003) or coordinate ascent variational inference (CAVI) (Bishop & Nasrabadi, 2006; Blei et al., 2017). The procedure consists of two parts:

At a given variational iteration $t$, we fix the posterior over parameters $q\left(\mathbf{\Theta}\right)$ to their value at the last iteration $q_{t-1}\left(\mathbf{\Theta}\right)$ (or if $t = 1$, to randomly-initialized values). With the parameter posterior fixed, we then update the posterior over latent variables by setting them equal to the solution that maximizes $\mathcal{L}(q)$, under $q_{t-1}\left(\mathbf{\Theta}\right)$:

$$q_t(\mathbf{X}_1, Z_1) \propto \exp\left\{ \mathbb{E}_{q_{t-1}(\mathbf{\Theta})} \left[ \ln p_{\boldsymbol{\Theta}}\left(Y, \mathbf{X}_1, Z_1 | \mathbf{X}_0\right) \right] \right\} \tag{7}$$

This update of the latents is also known as the 'variational E-step' due to its resemblance to the E-step in expectation maximization (Beal, 2003). The posterior over latent variables updated in the E-step of Equation (7) $q_t(\mathbf{X}_1, Z_1)$ is then used to update the posterior over parameters as

$$q_t\left(\mathbf{\Theta}\right) \propto \exp\left\{ \sum_{n=1}^{N} \mathbb{E}_{q_t\left(\mathbf{x}_1^n, z_1^n\right)} \left[ \ln p_{\boldsymbol{\Theta}}\left(y^n, \mathbf{x}_1^n, z_1^n | \mathbf{x}_0^n\right) \right] \right\} \tag{8}$$

This update of the parameter posterior is similarly known as the 'variational M-step' due to its resemblance to the step which maximizes the log likelihood with respect to the parameters in the E-M algorithm. More detailed expansions of these equations, including their functional forms, and the PG augmentation scheme needed to turn them into conditionally-conjugate updates, are given in Appendix A and Appendix B.

## 4 RESULTS

To evaluate the effectiveness of the CAVI-based approach, we compared it to other approximate inference algorithms, using several real and synthetic datasets. We compared CMNs fit with CAVI to the following three approaches:

**MLE** — We obtained point estimates for the parameters $\mathbf{\Theta}$ of the CMN using maximum-likelihood estimation (backpropagation to minimize the negative log likelihood).

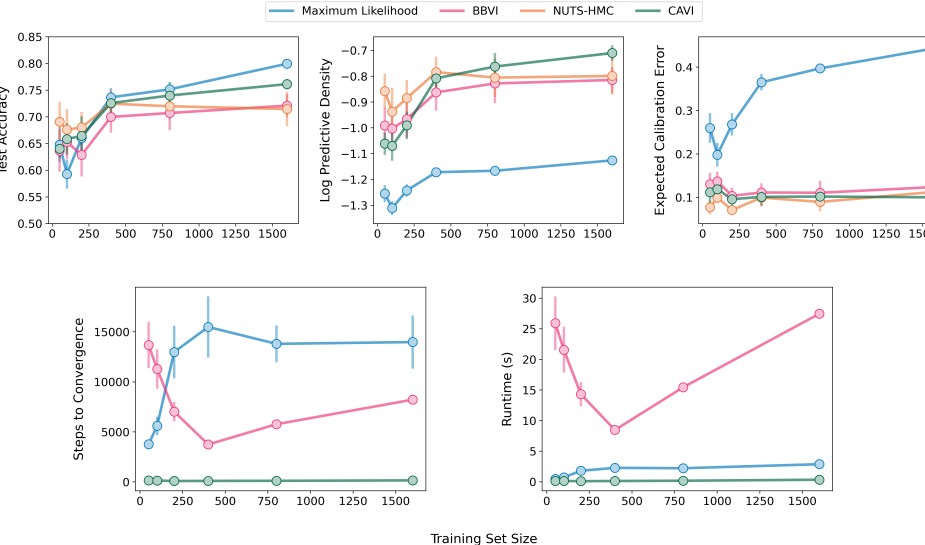

Figure 2: Performance and runtime results of the different inference algorithms on the 'Pinwheel' dataset. The standard deviation (vertical lines) of the performance metric is depicted together with the mean estimate (circles) over different runs. The top row of subplots show performance metrics across training set sizes: test accuracy (top left); log predictive density (top center), and expected calibration error (top right). The bottom row shows runtime metrics as a function of increasing training set size: the number of iterations required to achieve convergence (lower left); and the total runtime, estimated using the product of the number of iterations to convergence and the average cost (in seconds) for running one iteration (lower right). The number of iterations required for convergence was calculated by determining the number of gradient steps (or M steps, for CAVI) taken before the ELBO (or negative log likelihood, for MLE) reached 95% of its maximum value (see Appendix F for details on how these metrics were computed).

**NUTS-HMC** — The No-U-Turn Sampler (NUTS), an extension to Hamiltonian Monte Carlo (HMC) that incorporates adaptive step sizes (Hoffman et al., 2014). This provides samples from a posterior distribution over $\Theta$.

**BBVI** — Black-Box Variational Inference (BBVI) method (Ranganath et al., 2014). BBVI maximizes the evidence lower bound (ELBO) using stochastic estimation of its gradients with respect to variational parameters.

Appendix C contains details of the hyperparameters used for each inference algorithm.

### 4.1 COMPARISON ON SYNTHETIC DATASETS

We fit two-layer CMNs with different inference routines on two different synthetic datasets: the Pinwheels and the Waveform Domains (Breiman & Stone, 1988) datasets. The pinwheels dataset consists of multiple clusters arranged in a pinwheel pattern, posing a challenging task for mixture models (Johnson et al., 2016) due to the curved and elongated spatial distributions of the data. See Appendix D for the parameters we used to simulate the pinwheels dataset. Similarly, the Waveform Domains dataset consists of synthetic data generated to classify three different waveform patterns, where each class is described by 21 continuous attributes (Breiman & Stone, 1988).

We fit all inference methods while varying the training set size $N$ in order to study the robustness of each inference method's performance in the low data regime. For each inference method and value of $N$, we fit the model using the same batch of training data, but with 16 randomly-initialized models (different initial posterior samples or weights), and evaluated performance on the same fixed test set across values of $N$.

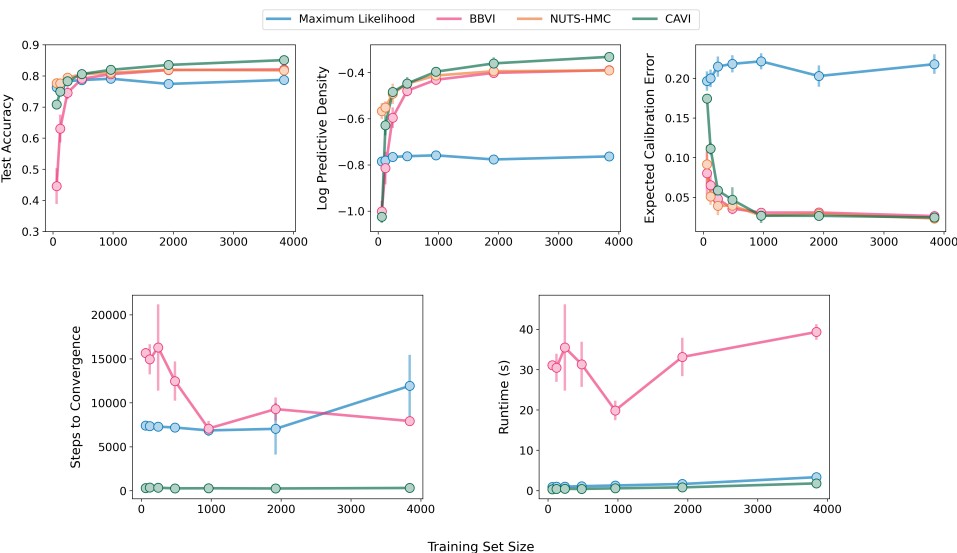

Figure 3: Performance and runtime results of the different models on the 'Waveform Domains' dataset. The waveforms dataset consists of synthetic data generated to classify three different waveform patterns. Each instance is described by 21 continuous attributes. See here for more information about the dataset. Descriptions of each subplot are same as in the Figure 2 legend.

We assess the performance of the different inference methods using three main metrics: predictive accuracy (Test Accuracy), log-predictive density (LPD), and expected calibration error (ECE). Log predictive density is a common measure of predictive accuracy for methods that output probabilities (Gelman et al., 2014), and expected calibration error measures how well a model's predictions are calibrated to the class probabilities observed in the data (Guo et al., 2017). In Figure 2 we show performance for the Pinwheels dataset and in Figure 3 for the Waveform dataset as a function of $N$. The CAVI-based approach achieves comparable log predictive density and calibration error to the other two Bayesian methods, which all outperform maximum likelihood estimation in LPD and ECE. This holds across training set sizes, indicating CAVI-CMN's high sample efficiency.

## 4.2 COMPARISON ON REAL-WORLD DATASETS

To further validate the performance of CAVI-CMN, we conducted experiments using 6 real-world classification datasets from the UCI Machine Learning Repository (Kelly et al., 2024). Table 1 summarizes the performance of the different algorithms on all 7 different UCI datasets (the Waveform domains dataset and the 6 real datasets), using the widely-applicable information criterion (WAIC) as a measure of performance. WAIC is an approximate estimate of leave-one-out cross-validation (Vehtari et al., 2017).

The CAVI-CMN approach consistently provided higher WAIC scores in comparison to the MLE algorithm, and with comparable magnitude to those computed using BBVI and NUTS. The results confirm that using fully conjugate priors within the CAVI framework, does not diminish the inference and the predictive performance of the algorithm, when compared to the state-of-the-art Bayesian methods like NUTS and BBVI. Importantly, CAVI-CMN offers substantial advantages in terms of computational efficiency as explored in the next section.

## 4.3 RUNTIME COMPARISON

The NUTS algorithm, although considered state-of-the-art in terms of inference robustness and accuracy (for well calibrated models (Gelman et al., 2020)), is notoriously difficult to apply to large-scale problems (Cobb & Jalaian, 2021). Hence, the preferred algorithm of choice for probabilistic machine learning applications have been methods grounded in variational inference, such as black-

Table 1: Comparison of widely-applicable information criterion (WAIC) for different methods evaluated on 7 different UCI datasets.

|  | Rice | Breast Cancer | Waveform | Vehicle Silh. | Banknote | Sonar | Iris |
|---|---|---|---|---|---|---|---|
| CAVI | -0.1820 | -0.0504 | -0.2921 | -0.3281 | -0.0206 | -0.1544 | -0.0747 |
| MLE | -0.3599 | -0.3133 | -0.5759 | -0.7437 | -0.3133 | -0.3133 | -0.5514 |
| NUTS | -0.1278 | -0.0324 | -0.3753 | -0.3767 | -0.0110 | -0.0306 | -0.0413 |
| BBVI | -0.1739 | -0.0763 | -0.3618 | -0.4154 | -0.0382 | -0.0583 | -0.1544 |

box variational inference (BBVI) (Ranganath et al., 2014) and stochastic variational inference (SVI) (Hoffman et al., 2013).

In this subsection, we analyze the runtime efficiency of the MLE and BBVI algorithms for CMN models, in comparison to a CAVI-based approach. The focus is on comparing the computation time as the number of model parameters increases. To study this, we varied the complexity of the CMN along the following dimensions: the dataset size used for training $N$, the number of linear experts in the mixture layer $K$, the dimensionality of the input space $d$, and the dimensionality of the latent space $h$. Note that in all runtime experiments, we did not measure NUTS' runtime because its poor scaling and speed is well-documented (Blei et al., 2017; Izmailov et al., 2021; Cobb & Jalaian, 2021) and for all but the smallest dataset sizes and model dimensionalities, it took already many times longer to run than any of the other methods we studied (MLE, BBVI, or CAVI).

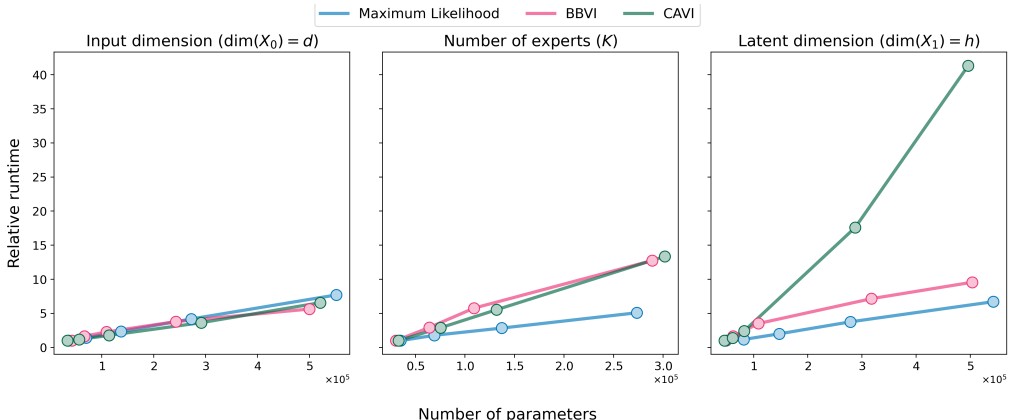

Figure 4: Relative scaling of fitting time in seconds for Maximum Likelihood, BBVI, and CAVI, as a function of the number of parameters. The number of parameters itself was manipulated in three illustrative ways: changing the input dimension $d$, changing the number of linear experts $K$ in the conditional mixture layer, and changing the dimensionality of the continuous latent variable $h$.

The runtime performance for varying dataset size $N$ is shown for the Pinwheels dataset in the bottom two subplots of Figure 2. This shows the total runtime in seconds, and steps until convergence for different algorithms. As expected, all algorithms exhibit an increase in runtime as $N$ increases (which also scales the number of parameters for BBVI and CAVI). However the rate of increase varies significantly across different algorithms, with CAVI-CMN approach showing the best scaling behavior.

Similarly, in Figure 4 we plot the relative runtime scaling of MLE, CAVI, and BBVI (proportional to the runtime of the least complex variant), as we increase the number of parameters along different dimensions of the model structure. Fitting CMNs with CAVI scales competitively with gradient-based methods like BBVI and MLE. However, the rightmost subplot of Figure 4 indicates that as we increase the dimensionality of the latent variable $\boldsymbol{X}_1$, CAVI-CMN scales more dramatically than the other two methods. This inherits from the computational overhead of matrix operations required by storing multivariate Gaussians posteriors over each continuous latent, i.e., $q(\boldsymbol{x}_1^n | z_1^n) = \mathcal{N}(\boldsymbol{x}_1^n; \boldsymbol{\mu}_1^n, \boldsymbol{\Sigma}_1^n)$. Running the CAVI algorithm involves operations (like matrix inversions and matrix-vector products) whose (naive) complexity is quadratic in matrix size. This explains the nonlinear scaling of runtime as a function of $h$, the dimension of $\boldsymbol{X}_1$. Various meth-

ods (like low-rank approximations to the covariance structure of $q(\boldsymbol{x}_1^n|z_1^n)$ and further factorization of $q(\boldsymbol{x}_1^n, z_1^n)$ into $q(\boldsymbol{x}_1^n)q(z_1^n)$) could help mitigate the scaling of CAVI-CMN's runtime as we increase the latent dimension $h$. These additions are fruitful avenue for future research into scaling CAVI-CMN to deeper (more layers) and wider (larger latent dimension) architectures.

In summary, both sets of runtime analyses (both absolute and relative) suggest CAVI-CMN may be an attractive alternative to BBVI suitable for large-scale and time-sensitive applications. Like BBVI, CAVI-CMN offers a variational Bayesian treatment of latent variables and parameters, while maintaining much faster absolute runtime and quicker convergence, due to the use of coordinate ascent to update the parameters of variational distributions, as opposed to the stochastic gradient updates used in BBVI.

## 5 CONCLUSION

We demonstrate that the CAVI-based approach for conditional mixture networks (CMN) significantly outperforms the traditional maximum likelihood estimation (MLE) based approach, in terms of predictive performance and calibration. The improvement in probabilistic performance over the MLE based approaches can be attributed to implicit regularisation via prior information, and proper handling of posterior uncertainty over latent states and parameters, leading to a better representation of the underlying data, reflected in improved calibration error and log predictive density, even in low data regimes.

One of the key advantages of the CAVI-based approach is its computational efficiency compared to the other Bayesian inference methods such as Black-Box variational inference (BBVI) and the No-U-turn sampler (NUTS). While NUTS can sample from the full joint posterior distribution, which maximizes performance in terms of inference quality, this comes at the expense of substantial computational resources, especially for high dimensional and complex models (Hoffman et al., 2013). Variational methods offer a scalable alternative to sampling-based inference in the form of methods like black-box variational inference (BBVI). Although BBVI is highly efficient in comparison to NUTS, it takes longer to converge and is slower than CAVI when applied to inference and learning in conditional mixture networks. Hence, we expect CAVI to be a more practical choice for large-scale application, especially when further combined with data mini-batching methods (Hoffman et al., 2013).

The UCI benchmark results show that CAVI-CMN algorithm achieves comparable performance to BBVI and NUTS in terms of predictive accuracy, log-predictive density and expected calibration error, while being significantly faster. This balance between predictive likelihood and calibration (jointly viewed as indicators of sample efficiency) is particularly important in real-world applications where robust prediction, reflective of underlying uncertainty, are crucial.

Furthermore, a straightforward mixture of linear components present in CMN, offers additional interoperability benefits. By using conditionally conjugate priors and a corresponding mean-field approximation over latent variables and model parameters, we facilitate easier interpretation of the model parameters and their uncertainties. This is particularly valuable in domains where understanding the underlying data-generating process is as important as the predictive performance, such as in healthcare, finance, and scientific research. Another important point is that the conjugate form of the CMN means that variational updates end up resembling sums of sufficient statistics computed from the data; this means the CAVI algorithm we described is readily amenable to online computation and mini-batching, where sufficient statistics can computed and summed on-the-fly to update model parameters in a streaming fashion (Hoffman et al., 2013; Broderick et al., 2013). This approach will become necessary when scaling CAVI-CMN to deeper (more than two-layer) models (Viroli & McLachlan, 2019) and larger datasets, where storing all the sufficient statistics of the data in memory becomes prohibitive.

Overall, these findings underscore the practical advantages of CAVI-CMN and highlight its promise as a new tool for fast probabilistic machine learning.

REPRODUCIBILITY STATEMENT

An author- and affiliation-anonymized version of the code for using CAVI and the other 3 methods to fit the CMN model on the pinwheel and 7 UCI datasets is available for download at this link. The main performance figures in the text can be reproduced by running each inference script and changing the `--train_size` parameter, while using the hyperparameters specified in the appendices (e.g., Appendix C.1).

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

## A  COORDINATE ASCENT VARIATIONAL INFERENCE FOR CONDITIONAL MIXTURE NETWORKS

In this section we expand the CAVI (or VBEM) equations provided in Equation (7) and Equation (8) of the main text to provide their detailed functional form.

Recall the ELBO used for CAVI-CMN:

$$\mathcal{L}(q) = \mathbb{E}_{q(\boldsymbol{X}_1, \boldsymbol{Z}_1)q(\boldsymbol{\Theta})} \left[ \sum_{n=1}^{N} \ln \frac{p_{\boldsymbol{\Theta}}\left(y^n, \boldsymbol{x}_1^n, z_1^n | \boldsymbol{x}_0^n\right)}{q\left(z_1^n\right) q\left(\boldsymbol{x}_1^n | z_1^n\right)} \right] + \mathbb{E}_{q(\boldsymbol{\Theta})} \left[ \ln \frac{p\left(\boldsymbol{\beta}_1\right) p\left(\boldsymbol{\beta}_0\right) p\left(\boldsymbol{\lambda}_1\right)}{q\left(\boldsymbol{\beta}_1\right) q\left(\boldsymbol{\beta}_0\right) q\left(\boldsymbol{\lambda}_1\right)} \right] \quad (9)$$

Coordinate ascent variational inference entails maximizing $\mathcal{L}(q) =$ using an iterative update scheme for the parameters of the approximate posterior, which factorizes into the posterior over parameters $q(\boldsymbol{\Theta})$ and the posterior over latent variables $q(\mathbf{X}_1, Z_1)$. The procedure consists of two parts:

First, we fix the posterior over the parameters to its value at the previous iteration (or randomly-initialized values, if at the first iteration). Given the posterior over parameters, we update the posterior over latent variables (variational E-step) as

$$q_t\left(\boldsymbol{x}_1^n | z_1^n\right) \propto \exp \left\{ \mathbb{E}_{q_{t-1}(\boldsymbol{\beta}_1)q_{t-1}(\boldsymbol{\lambda}_1)} \left[ \ln p_{\boldsymbol{\beta}_1}\left(y^n | \boldsymbol{x}_1^n\right) + \ln p_{\boldsymbol{\lambda}_1}\left(\boldsymbol{x}_1^n | \boldsymbol{x}_0^n, z_1^n\right) \right] \right\}$$

$$q_t\left(z_1^n\right) \propto \exp \left\{ \mathbb{E}_{q_{t-1}(\boldsymbol{\Theta})} \left[ \left\langle \ln p_{\boldsymbol{\beta}_1, \boldsymbol{\lambda}_1}\left(y^n, \boldsymbol{x}_1^n | \boldsymbol{x}_0^n, z_1^n\right) \right\rangle_{q_t\left(\boldsymbol{x}_1^n | z_1^n\right)} + \ln p_{\boldsymbol{\beta}_0}\left(z_1^n | \boldsymbol{x}_0^n\right) \right] \right\} \quad (10)$$

Second, the posterior over latents that was updated in the E-step, is used to update the posterior over parameters (variational M-step) as

$$q_t\left(\boldsymbol{\beta}_1\right) \propto \exp \left\{ \sum_{n=1}^{N} \mathbb{E}_{q_t(\boldsymbol{x}_1^n, z_1^n)} \left[ \ln p_{\boldsymbol{\beta}_1}\left(y^n | \boldsymbol{x}_1^n\right) \right] \right\}$$

$$q_t\left(\boldsymbol{\beta}_0\right) \propto \exp \left\{ \sum_{n=1}^{N} \mathbb{E}_{q_t(z_1^n)} \left[ \ln p_{\boldsymbol{\beta}_1}\left(z_1^n | \boldsymbol{x}_0^n\right) \right] \right\} \quad (11)$$

$$q_t\left(\boldsymbol{\lambda}_1\right) \propto \exp \left\{ \sum_{n=1}^{N} \mathbb{E}_{q_t(\boldsymbol{x}_1^n, z_1^n)} \left[ \ln p_{\boldsymbol{\lambda}_1}\left(\boldsymbol{x}_1^n | z_1^n, \boldsymbol{x}_0^n\right) \right] \right\}$$

In the variational inference literature, the distinction between latents and parameters is often described in terms of 'local' vs 'global' latent variables (Hoffman et al., 2013), where local variables are datapoint-specific, and global variables are shared across datapoints. To detail the form of the updates to the parameters of the linear experts in Equation (11), i.e. $q_t(\boldsymbol{\lambda}_1) = q_t(\boldsymbol{A}_{1:K}, \boldsymbol{\Sigma}_{1:K}^{-1})$, first we note the form of the approximate posteriors over the latent variables $q(\boldsymbol{X}_1, Z_1)$:

$$q\left(\boldsymbol{X}_1 | Z_1\right) = \prod_{n=1}^{N} \prod_{k=1}^{K} \mathcal{N}(\boldsymbol{x}_1^n; \boldsymbol{\mu}_{k,1}^n, \boldsymbol{\Sigma}_{k,1}^n)$$

$$q\left(Z_1\right) = \prod_{n=1}^{N} \text{Cat}(z_1^n; \boldsymbol{\gamma}^n) \quad (12)$$

The update to the $k^{\text{th}}$ expert's parameters $q(\boldsymbol{A}_k, \boldsymbol{\Sigma}_k^{-1})$ can written in terms of weighted updates to the Matrix Normal Gamma's canonical parameters $\boldsymbol{M}_k, \boldsymbol{V}_k, a_k$ and $b_k$, where the weights are provided by the sufficient statistics of $\{q\left(\boldsymbol{x}_1^1 | z_1^1 = k\right), q\left(\boldsymbol{x}_1^2 | z_1^2 = k\right), \dots, q\left(\boldsymbol{x}_1^N | z_1^N = k\right)\}$:

$$\boldsymbol{V}_k^{-1} = \boldsymbol{V}_{k,0}^{-1} + \sum_{n=1}^N \gamma_k^n \boldsymbol{x}_0^n \left(\boldsymbol{x}_0^n\right)^\top$$

$$\boldsymbol{M}_k = \left(\boldsymbol{M}_{k,0}\boldsymbol{V}_{k,0}^{-1} + \sum_{n=1}^N \gamma_k^n \boldsymbol{\mu}_{k,1}^n \left(\boldsymbol{x}_0^n\right)^\top\right) \boldsymbol{V}_k$$

$$a_k = a_{k,0} + \frac{\sum_{n=1}^N \gamma_k^n}{2}$$

$$b_{i,k} = b_{i,k,0} + \frac{1}{2}\left(\sum_{n=1}^N \gamma_k^n \left[\boldsymbol{\Sigma}_{k,1}^n + \boldsymbol{\mu}_{k,1}^n(\boldsymbol{\mu}_{k,1}^n)^\top\right]_{ii} - \left[\boldsymbol{M}_k\boldsymbol{V}_k^{-1}\boldsymbol{M}_k^T\right]_{ii} + \left[\boldsymbol{M}_{k,0}\boldsymbol{V}_{k,0}^{-1}\boldsymbol{M}_{k,0}^T\right]_{ii}\right)$$

$$(13)$$

where the notation $[\cdot]_{ii}$ selects the $i^{\text{th}}$ element of the diagonal of the matrix in the brackets.

However, the update equations described in Equation (10) and in the first two lines of Equation (11) for $q(\boldsymbol{\beta}_0), q(\boldsymbol{\beta}_1)$ are not computationally tractable without an additional approximation, known as Pólya-Gamma augmentation of the multinomial distribution. The full details of the augmentation procedure are described below in Appendix B. Here we will briefly sketch the main steps and describe the high level, augmented update equations. The Pólya-Gamma augmentation introduces datapoint-specific auxiliary variables $(\boldsymbol{\omega}_1^n, \boldsymbol{\omega}_0^n)$, that help us transform the log-probability of the multinomial distribution into a quadratic function (Polson et al., 2013; Linderman et al., 2015) over coefficients $(\boldsymbol{\beta}_1, \boldsymbol{\beta}_0)$, and latents $\boldsymbol{x}_1^n$. This quadratic form enables tractable update of $q\left(\boldsymbol{x}_1^n|z_1^n\right)$ in the form of a multivariate normal distribution, and a tractable updating of posteriors over coefficients $q\left(\boldsymbol{\beta}_1\right)$ and $q\left(\boldsymbol{\beta}_0\right)$.

With the introduction of the auxiliary variables the variational expectation and maximisation steps are expressed as

Update latents ('E-step')

$$q_t\left(\boldsymbol{x}_1^n|z_1^n\right) \propto \exp\left\{\mathbb{E}_{q_{t-1}(\boldsymbol{\beta}_1)q_{t-1}(\boldsymbol{\lambda}_1)}\left[\left\langle l\left(y^n, \boldsymbol{x}_1^n, \boldsymbol{\omega}_1^n, \boldsymbol{\beta}_1\right)\right\rangle_{q_{t-1}\left(\boldsymbol{\omega}_1^n|y^n\right)} + \ln p_{\boldsymbol{\lambda}_1}\left(\boldsymbol{x}_1^n|\boldsymbol{x}_0^n, z_1^n\right)\right]\right\}$$

$$q_t\left(\boldsymbol{\omega}_1|y^n\right) \propto p\left(\boldsymbol{\omega}_1^n|y_n\right)\exp\left\{\mathbb{E}_{q_{t-1}(\boldsymbol{\beta}_1)q_t\left(\boldsymbol{x}_1^n|z_1^n\right)}\left[l\left(y^n, \boldsymbol{x}_1^n, \boldsymbol{\omega}_1^n, \boldsymbol{\beta}_1\right)\right]\right\}$$

$$q_t\left(\boldsymbol{\omega}_0|z_1^n\right) \propto p\left(\boldsymbol{\omega}_0^n|z_1^n\right)\exp\left\{\mathbb{E}_{q_{t-1}(\boldsymbol{\beta}_0)}\left[l\left(z_1^n, \boldsymbol{x}_0^n, \boldsymbol{\omega}_0^n, \boldsymbol{\beta}_0\right)\right]\right\}$$

$$q_t\left(z_1^n\right) \propto \exp\left\{\mathbb{E}_{q_{t-1}(\boldsymbol{\Theta})}\left[\bar{l}_{z_1^n,t}\left(y^n, \boldsymbol{\beta}_1\right) + R_{z_1^n,t}\left(\boldsymbol{x}_0^n, \boldsymbol{\lambda}_1\right) + \bar{l}_t\left(z_1^n, \boldsymbol{x}_0^n, \boldsymbol{\beta}_0\right)\right]\right\}$$

Update parameters ('M-step')

$$q_t\left(\boldsymbol{\beta}_1\right) \propto \exp\left\{\sum_{n=1}^N \mathbb{E}_{q_t\left(\boldsymbol{x}_1^n, z_1^n\right)q_t\left(\boldsymbol{\omega}_1^n|y^n\right)}\left[l(y^n, \boldsymbol{x}_1^n, \boldsymbol{\beta}_1, \boldsymbol{\omega}_1^n)\right]\right\}$$

$$q_t\left(\boldsymbol{\beta}_0\right) \propto \exp\left\{\sum_{n=1}^N \mathbb{E}_{q_t\left(z_1^n\right)q_t\left(\boldsymbol{\omega}_0^n|z_1^n\right)}\left[l(z_1^n, \boldsymbol{x}_0^n, \boldsymbol{\beta}_0, \boldsymbol{\omega}_0^n)\right]\right\}$$

$$(14)$$

where we skipped the terms whose form did not change. $R_{z_1^n,t}\left(\boldsymbol{x}_0^n, \boldsymbol{\lambda}_1\right)$ reflects a contribution to $q(z_1^n)$ that depends on the expected log partition of the linear (Matrix Normal Gamma) likelihood $p_{\boldsymbol{\lambda}_1}(\boldsymbol{x}_1^n|\boldsymbol{x}_0^n, z_1^n)$. Note that the updates to each subset of posteriors (latents or parameters) have an analytic form due to the conditional conjugacy of the model. Importantly, both priors and posterior of the auxiliary variables are Pólya-Gamma distributed (Polson et al., 2013).

Finally, in the above update equations, we have replaced instances of the multinomial distribution $p(z|\boldsymbol{x}, \boldsymbol{\beta})$ with its augmented form $p(\omega|z) e^{l(z,\boldsymbol{x},\boldsymbol{\omega},\boldsymbol{\beta})}$ where the function $l(\cdot)$ is quadratic with respect to the coefficients $\boldsymbol{\beta}$ and the input variables $\boldsymbol{x}$, leading to tractable update equations.

# B  VARIATIONAL BAYESIAN MULTINOMIAL LOGISTIC REGRESSION

In this section, we focus on a single multinomial logistic regression model (not in the context of the CMN), but the ensuing variational update scheme derived in Appendix B.4 is applied in practice to both the gating network's parameters $\boldsymbol{\beta}_0$ as well as those of the final output likelihood for the class label $\boldsymbol{\beta}_1$.

## B.1  STICK-BREAKING REPARAMETERIZATION OF A MULTINOMIAL DISTRIBUTION

Multinomial logistic regression considers the probability that an outcome variable $y$ belongs to one of $K$ mutually-exclusive classes or categories. The probability of $y$ belonging to the $k^{\text{th}}$ class is given by the categorical likelihood:

$$p(y = k|\boldsymbol{x}, \boldsymbol{\beta}) = p_k \tag{15}$$

The problem of multinomial logistic regression is to identify or estimate the values of regression coefficients $\boldsymbol{\beta}$ that explain the relationship between some dataset of given continuous input regressors $\boldsymbol{X} = (\boldsymbol{x}^1, \boldsymbol{x}^2, \ldots, \boldsymbol{x}^N)$ and corresponding categorical labels $Y = (y^1, y^2, \ldots, y^N), y^n \in 1, 2, \ldots, K$.

We can use a stick-breaking construction to parameterize the likelihood over $y$ using a set of $K - 1$ stick-breaking coefficients: $\boldsymbol{\pi} = (\pi_1, \ldots, \pi_{K-1})$. Each coefficient is parameterized with an input regressor $\boldsymbol{x}$, and a corresponding set of regression weights $\boldsymbol{\beta}_j$. Stick-breaking coefficient $\pi_j$ is then given by a sigmoid transform of the product of the regression weights and the input regressors:

$$\pi_j = \sigma\left(\boldsymbol{\beta}_j\left[\boldsymbol{x}; 1\right]\right) ,$$
$$\text{where } \sigma\left(\boldsymbol{\beta}_j\left[\boldsymbol{x}; 1\right]\right) = \frac{1}{1 + \exp\left\{-\boldsymbol{\beta}_j\left[\boldsymbol{x}; 1\right]\right\}} ,$$
$$\text{and } \boldsymbol{\beta}_j\left[\boldsymbol{x}; 1\right] = \sum_{i=1}^{d} w_{j,i} x_i + a_j. \tag{16}$$

The outcome likelihood is then obtained via stick breaking transform[1] as follows

$$p_k = \pi_K \prod_{j=1}^{K-1}(1 - \pi_j) = \sigma\left(\boldsymbol{\beta}_K\left[\boldsymbol{x}; 1\right]\right) \prod_{j=1}^{K-1}\left(1 - \sigma\left(\boldsymbol{\beta}_j\left[\boldsymbol{x}; 1\right]\right)\right) = \prod_{j=1}^{K-1}\frac{\exp\left\{\boldsymbol{\beta}_j\left[\boldsymbol{x}; 1\right]\right\}}{1 + \exp\left\{\boldsymbol{\beta}_j\left[\boldsymbol{x}; 1\right]\right\}} \tag{17}$$

where $\pi_K = 1$, and $\boldsymbol{\beta}_K = \vec{0}$.

Finally, we can express the likelihood in the form of a Categorical distribution as

$$\text{Cat}(y; \boldsymbol{x}, \boldsymbol{\beta}) = \prod_{k=1}^{K-1}\frac{\left(\exp\left\{\boldsymbol{\beta}_k\left[\boldsymbol{x}; 1\right]\right\}\right)^{\delta_{k,y}}}{\left(1 + \exp\left\{\boldsymbol{\beta}_k\left[\boldsymbol{x}; 1\right]\right\}\right)^{N_{k,y}}} . \tag{18}$$

where $N_{k,y} = 1$ for $k \leq y$, and $N_{k,y} = 0$ otherwise (or $N_{k,y} = 1 - \sum_{j=1}^{k-1}\delta_{j,y}$), and $\delta_{k,y} = 1$ for $k = y$ and is zero otherwise.

---

[1]This blog post has helpful discussion on the stick-breaking form of the multinomial logistic likelihood and provides more intuition behind its functional form.

## B.2 Pólya-Gamma augmentation

The *Pólya-Gamma augmentation* scheme (Polson et al., 2013; Linderman et al., 2015; Durante & Rigon, 2019) is defined as

$$\frac{\left(e^{\psi}\right)^a}{\left(1 + e^{\psi}\right)^b} = 2^{-b} e^{\kappa\psi} \int_0^{\infty} e^{-\omega\psi^2/2} p(\omega)\, \mathrm{d}\omega \tag{19}$$

where $\kappa = a - b/2$ and $p\left(\omega|b,0\right)$ is the density of the Pólya-Gamma distribution $PG(b,0)$ which does not depend on $\psi$. The useful properties of the Pólya-Gamma are the exponential tilting property expressed as

$$PG(\omega; b, \psi) = \frac{e^{-\omega\psi^2/2} PG(\omega; b, 0)}{\mathbb{E}\left[e^{-\omega\psi^2/2}\right]} \tag{20}$$

the expected value of $\omega$, and $e^{-\omega\psi^2/2}$ given as

$$\mathbb{E}\left[\omega\right] = \int_o^{\infty} \omega PG(\omega; b, \psi)\, \mathrm{d}\omega = \frac{b}{2\psi} \tanh\left(\frac{\psi}{2}\right),$$

$$\mathbb{E}\left[e^{-\omega\psi^2/2}\right] = \cosh^{-b}\left(\frac{\psi}{2}\right) \tag{21}$$

and the Kulback-Leibler divergence between $q\left(\omega\right) = PG(\omega; b, \psi)$ and $p\left(\omega\right) = PG(\omega; b, 0)$ obtained as

$$D_{KL}\left[q\left(\omega\right)||p\left(\omega\right)\right] = -\mathbb{E}\left[\omega\right]\frac{\psi^2}{2} + b\ln\cosh\left(\frac{\psi}{2}\right) = -\frac{b\psi}{4}\tanh\left(\frac{\psi}{2}\right) + b\ln\cosh\left(\frac{\psi}{2}\right). \tag{22}$$

We can express the likelihood function in Equation (18) using the augmentation as

$$p(y, \boldsymbol{\omega}|\boldsymbol{\psi}) = p\left(y|\boldsymbol{\psi}\right) p\left(\boldsymbol{\omega}|y, \boldsymbol{\psi}\right) = \prod_{k=1}^{K-1} 2^{-b_{k,y}} e^{\kappa_{k,y}\psi_k - \omega_k\psi_k^2/2} \mathrm{PG}(\omega_k; b_{k,y}, 0)$$

$$p\left(y|\boldsymbol{\psi}\right) = \prod_{k=1}^{K-1} 2^{-b_{k,y}} e^{\kappa_{k,y}\psi_k} \int_0^{\infty} e^{-\omega_k\psi_k^2/2} \mathrm{PG}(\omega_k; b_{k,y}, 0)\, \mathrm{d}\omega_k \tag{23}$$

$$p\left(\boldsymbol{\omega}|y, \boldsymbol{\psi}\right) = \prod_{k=1}^{K-1} \mathrm{PG}\left(\omega_k; b_{k,y}, \psi_k\right)$$

where $b_{k,y} \equiv N_{k,y}$, $\kappa_{k,y} = \delta_{k,y} - N_{k,y}/2$, and $\psi_k = \boldsymbol{\beta_k}\left[\boldsymbol{x}; 1\right]$. Given a prior distribution $p\left(\boldsymbol{\psi}\right) = p\left(\boldsymbol{\beta}\right) p\left(\boldsymbol{x}\right)$, we can write the joint $p\left(y, \boldsymbol{\omega}, \boldsymbol{\psi}\right)$ as

$$p\left(y, \boldsymbol{\omega}, \boldsymbol{\psi}\right) = p\left(\boldsymbol{\omega}|y\right) p\left(\boldsymbol{\psi}\right) e^{l(y, \boldsymbol{\psi}, \boldsymbol{\omega})},$$

$$l\left(y, \boldsymbol{\psi}, \boldsymbol{\omega}\right) = \sum_{k=1}^{K-1} l_k\left(y, \psi_k, \omega_k\right), \tag{24}$$

$$l_k\left(y, \psi_k, \omega_k\right) = \kappa_{y,k}\psi_k - b_{y,k}\ln 2 - \omega_k\psi_k^2/2.$$

## B.3 Evidence lower-bound

Given a set of observations $\boldsymbol{\mathcal{D}} = \left(y^1, \ldots, y^N\right)$ the augmented joint distribution can be expressed as

$$p\left(\boldsymbol{\mathcal{D}}, \boldsymbol{\Omega}, \boldsymbol{X}, \boldsymbol{\beta}\right) = p\left(\boldsymbol{\beta}\right) \prod_{n=1}^{N} p\left(\boldsymbol{x}^n\right) p\left(\boldsymbol{\omega}^n|y^n\right) e^{l(y^n, \boldsymbol{\psi}^n, \boldsymbol{\omega}^n)}$$

We can express the evidence lower-bound (ELBO) as

$$
\begin{aligned}
\mathcal{L}(q) &= \mathbb{E}_{q(\boldsymbol{\Omega})q(\boldsymbol{X})q(\boldsymbol{\beta})} \left[ -\ln q\left(\boldsymbol{\beta}\right) + \sum_{n=1}^{N} \ln \frac{p\left(y^n, \boldsymbol{\psi}^n, \boldsymbol{\omega}^n\right)}{q\left(\boldsymbol{\omega}^n\right) q\left(\boldsymbol{x}^n\right)} \right] \\
&= \mathbb{E}_{q(\boldsymbol{\Omega})q(\boldsymbol{X})q(\boldsymbol{\beta})} \left[ \ln \frac{p\left(\boldsymbol{\beta}\right)}{q\left(\boldsymbol{\beta}\right)} + \sum_{n=1}^{N} l\left(y^n, \boldsymbol{\psi}^n, \boldsymbol{\omega}^n\right) + \ln \frac{p\left(\boldsymbol{\omega}^n|y^n\right)}{q\left(\boldsymbol{\omega}^n\right)} + \ln \frac{p\left(\boldsymbol{x}^n\right)}{q\left(\boldsymbol{x}^n\right)} \right] \\
&\geq \ln p\left(\boldsymbol{\mathcal{D}}\right)
\end{aligned}
\tag{25}
$$

where we use the following forms for the approximate posterior

$$
\begin{aligned}
q\left(\boldsymbol{\Omega}|Y\right) &= \prod_{n=1}^{N} q\left(\boldsymbol{\omega}^n|y^n\right) = \prod_{n=1}^{N} \prod_{k=1}^{K-1} PG\left(b_{k,y^n}, \xi_{k,n}\right) , \\
q\left(\boldsymbol{X}\right) &= \prod_{n=1}^{N} q\left(\boldsymbol{x}^n\right) = \prod_{n=1}^{N} \mathcal{N}\left(\boldsymbol{x}^n; \boldsymbol{\mu}^n, \boldsymbol{\Sigma}^n\right) , \\
q\left(\boldsymbol{\beta}\right) &= \prod_{k=1}^{K-1} \mathcal{N}\left(\boldsymbol{\beta}_k; \boldsymbol{\mu}_k, \boldsymbol{\Sigma}_k\right) .
\end{aligned}
\tag{26}
$$

### B.4 Coordinate ascent variational inference

The mean-field assumption in Equation (26) allows the implementation of a simple CAVI algorithm (Wainwright et al., 2008; Beal, 2003; Hoffman et al., 2013; Blei et al., 2017) which sequentially maximizes the evidence lower bound in Equation (25) with respect to each factor in $q\left(\boldsymbol{\Omega}|Y\right) q\left(\boldsymbol{X}\right) q\left(\boldsymbol{\beta}\right)$, via the following updates:

Update to latents ('E-step')

$$
q^{(t,l)}\left(\boldsymbol{x}^n\right) \propto p\left(\boldsymbol{x}^n\right) \exp\left\{ \mathbb{E}_{q^{(t-1)}(\boldsymbol{\beta})q^{(t,l-1)}(\boldsymbol{\omega}^n)} \left[ l\left(y^n, \boldsymbol{\psi}^n, \boldsymbol{\omega}^n\right) \right] \right\}
$$

$$
q^{(t,l)}\left(\omega_k^n|y^n\right) \propto p\left(\omega_k^n|y^n\right) \exp\left\{ \mathbb{E}_{q^{(t-1)}(\boldsymbol{\beta})q^{(t,l)}(\boldsymbol{x}^n)} \left[ l_k\left(y^n, \psi_k^n, \omega_k^n\right) \right] \right\}
$$

$$
\forall n \in \{1, \ldots, N\}, \text{ and for } q^{(t,0)}\left(\boldsymbol{\omega}^n|y^n\right) = q^{(t-1,L)}\left(\boldsymbol{\omega}^n|y^n\right) \tag{27}
$$

Update to parameters ('M-step')

$$
q^{(t)}\left(\boldsymbol{\beta}_k\right) \propto \exp\left\{ \sum_{n=1}^{N} \mathbb{E}_{q^{(t)}(\boldsymbol{x}^n)q^{(t)}(\boldsymbol{\omega}^n|y^n)} \left[ l\left(y^n, \boldsymbol{\psi}^n, \boldsymbol{\omega}^n\right) \right] \right\}
$$

at each iteration $t$, and multiple local iteration $l$ during the variational expectation step—until the convergence of the ELBO.

Specifically, the update equations for the parameters of the latents (the 'E-step') are:

$$
q^{(t,l)}\left(\boldsymbol{x}^n\right) \propto \mathcal{N}\left(\boldsymbol{x}^n; 0, -2\boldsymbol{\lambda}_{2,0}\right) \exp\left\{ \sum_{k=1}^{K} \kappa_{k,y^n} \operatorname{Tr}\left(\boldsymbol{\mu}_k^{(t-1)} [\boldsymbol{x}^n;1]^T\right) - \frac{\langle\omega_k\rangle}{2} \operatorname{Tr}\left(\boldsymbol{M}_k^{(t-1)} [\boldsymbol{x}^n;1][\boldsymbol{x}^n;1]^T\right) \right\}
$$

$$
\boldsymbol{\lambda}_1^{(n,t,l)} = \sum_{k=1}^{K-1} \left\{ \kappa_{k,y^n} \left[\boldsymbol{\mu}_k^{(t-1)}\right]_{1:D} - \langle\omega_k^n\rangle_{t,l-1} \left[\boldsymbol{M}_k^{(t-1)}\right]_{D+1,1:D} \right\}
$$

$$
\boldsymbol{\lambda}_2^{(n,t,l)} = \boldsymbol{\lambda}_{2,0} - \frac{1}{2} \sum_{k=1}^{K-1} \langle\omega_k^n\rangle_{t,l-1} \left[\boldsymbol{M}_k\right]_{1:D,1:D}
$$

$$
\boldsymbol{M}_k^{(t-1)} = \boldsymbol{\Sigma}_k^{(t-1)} + \boldsymbol{\mu}_k^{(t-1)} \left[\boldsymbol{\mu}_k^{(t-1)}\right]^T
$$

$$
\tag{28}
$$

and

$$q^{(t,l)}\left(\omega_k^n | y^n\right) \propto e^{-\omega_k^n \langle \psi_k^2 \rangle / 2} PG\left(\omega_k^n; b_{k,y^n}, 0\right)$$

$$\xi_k^n = \sqrt{\mathbb{E}_{q^{(t-1)}(\boldsymbol{\beta})q^{(t,l)}(\boldsymbol{x}^n)}\left[\psi_k^2\right]}$$

$$\xi_k^n = \sqrt{\mathrm{Tr}\left(\boldsymbol{M}_k^{(t-1)} \hat{\boldsymbol{M}}^{(n,t,l)}\right)} \tag{29}$$

where $\hat{\boldsymbol{M}}^{(n,t,l)} = \begin{pmatrix} \boldsymbol{M}^{(n,t,l)} & \boldsymbol{\mu}^{(n,t,l)} \\ \left[\boldsymbol{\mu}^{(n,t,l)}\right]^T & 1 \end{pmatrix}$ , and $\boldsymbol{M}^{(n,t,l)} = \boldsymbol{\Sigma}^{(n,t,l)} + \boldsymbol{\mu}^{(n,t,l)}\left[\boldsymbol{\mu}^{(n,t,l)}\right]^T$ .

Similarly, for the parameter updates ('M-step') we get

$$q^{(t)}\left(\boldsymbol{\beta}_k\right) \propto \mathcal{N}\left(\boldsymbol{\beta}_k; 0, -2\boldsymbol{\lambda}'_{2,0}\right) \exp\left\{\sum_{n=1}^N \kappa_{k,y^n} \mathrm{Tr}\left(\hat{\boldsymbol{\mu}}^{(n,t)} \boldsymbol{\beta}_k^T\right) - \frac{\langle \omega_k \rangle_t^n}{2} \mathrm{Tr}\left(\hat{\boldsymbol{M}}_i^{(t)} \boldsymbol{\beta}_k \boldsymbol{\beta}_k^T\right)\right\}$$

$$\boldsymbol{\lambda}_{k,1}^{(t)} = \sum_i \kappa_{k,y^n} \hat{\boldsymbol{\mu}}^{(n,t)} \tag{30}$$

$$\boldsymbol{\lambda}_{k,2}^{(t)} = \boldsymbol{\lambda}'_{2,0} - \frac{1}{4} \sum_{n=1}^N \frac{b_{k,y^n}}{\xi_k^{(n,t)}} \tanh\left(\frac{\xi_k^{(n,t)}}{2}\right) \hat{\boldsymbol{M}}^{(n,t)}$$

where $\hat{\boldsymbol{\mu}}^{(n,t)} = \left[\boldsymbol{\mu}^{(n,t)}; 1\right]$.

## C  HYPERPARAMETERS

### C.1  COMMON HYPERPARAMETERS

For the Bayesian methods (CAVI, NUTS, and BBVI), we used the same form for the CMN priors (see Equation (3) for their parameterization) and fixed the prior parameters to the following values, used for all datasets: $v_0 = 10$, $a_0 = 2$, $b_0 = 1$, $\sigma_0, \sigma_1 = 5$. For all datasets, we fixed the dimension of the continuous latent $\boldsymbol{x}_1$ to be $h = L - 1$, where $L$ is the number of classes. For the Pinwheels dataset (see Appendix D.1 below), we set the number of linear experts (and hence the dimension of the discrete latent $\boldsymbol{z}_1$) at $K = 10$, while for all other datasets we used $K = 20$.

### C.2  MAXIMUM LIKELIHOOD ESTIMATION

For gradient-based optimization of the loss function (the negative log likelihood), we used the AdaBelief optimizer with parameters set to its default values as introduced in Zhuang et al. (2020) ($\alpha = 1e-3$, $\beta_1 = 0.9$, $\beta_2 = 0.999$), and ran the optimization for $20,000$ steps. This implements deterministic gradient descent, not stochastic gradient descent, because we fit the model in 'full-batch' mode, i.e., without splitting the data into mini-batches and updating model parameters using noisy gradient estimates.

### C.3  NO U-TURN SAMPLER

Markov Chain Monte Carlo converges in distribution to samples from a target distribution, so for this method we obtain samples from a joint distribution $p(\boldsymbol{A}_{1:K}, \boldsymbol{\Sigma}_{1:K}^{-1}, \boldsymbol{\beta}_0, \boldsymbol{\beta}_1 | Y, \boldsymbol{X}_0)$ that approximate the true posterior. We used 800 warm-up steps, 16 independent chains, and 64 samples for each chain.

### C.4  BLACK BOX VARIATIONAL INFERENCE

While BBVI does not require conjugate relationships in the generative model, we use the same CMN model and variational distributions as we use for CAVI-CMN, in order to ensure fair comparison. For stochastic optimization, we used the AdaBelief optimizer with learning rate $\alpha = 5e-3$ $\beta_1 = 0.9$,

$\beta_2 = 0.999$, used 8 samples to estimate the ELBO gradient (the `num_particles` argument of the `Trace_ELBO()` class), and ran the optimizer for $20,000$ steps).

## D    DATASET DESCRIPTIONS

We fit all inference methods using different training set sizes, where each next training set was twice as large as the previous. For each training size, we used the same test-set to evaluate performance. The test set was ensured to have the same relative class frequencies as in the training set(s). For each inference method and examples set size, we fit using the same batch of training data, but with 16 randomly-initialized models (different initial posterior samples or parameters).

### D.1    PINWHEELS DATASET

The pinwheels dataset is a synthetic dataset designed to test a model's ability to handle nonlinear decision boundaries and data with non-Gaussian densities (Johnson et al., 2016). The structure of the pinwheels dataset is determined by 4 parameters: the number of clusters or distinct spirals; the angular deviation, which defines how far the spiralling clusters deviate from the origin; the tangential deviation, which defines the noise variance of 2-D points within each cluster; and the angular rate, which determines the curvature of each spiral. For evaluating the four methods (CAVI-CMN, MLE, BBVI, and NUTS) on the synthetic pinwheels dataset, we generated a dataset with 5 clusters, with an angular deviation of 0.7, tangential deviation of 0.3 and angular rate of 0.2. We selected these values by looking at the maximum achieved test accuracy across all the methods for different parameter combinations and tried to upper-bound it 80%, which provides a low enough signal-to-noise ratio to be able to meaningfully show differences in probabilistic metrics like calibration and WAIC. For pinwheels, we trained using train sizes 50 to 1600, doubling the number of training examples at each successive training set size. We tested using 500 held-out test examples generated using the same parameters as used for the training set(s).

### D.2    WAVEFORM DOMAINS DATASET

The Waveform Domains dataset consists of synthetic data generated to classify three different waveform patterns, where each class is described by 21 continuous attributes (Breiman & Stone, 1988). For waveform domains, we fit each model on train sizes ranging from 60 to 3840 examples, and tested on a held-out size of 1160 examples. See here for more information about the dataset.

### D.3    VEHICLE SILHOUETTES DATASET

This dataset involves classifying vehicle silhouettes into one of four types (bus, van, or two car models) based on features extracted from 2D images captured at various angles (Mowforth & Shepherd). We fit each model on train sizes ranging from 20 to 650 examples, and tested on a held-out size of 205 examples. See here for more information about the dataset.

### D.4    RICE DATASET

The Rice dataset contains measurements related to the classification of rice varieties, specifically Cammeo and Osmancik (mis, 2019). We fit each model on train sizes ranging from 40 to 2560 examples, and tested on a held-out size of 1250. See here for more information about the dataset.

### D.5    BREAST CANCER DATASET

The 'Breast Cancer Diagnosis' dataset (Wolberg et al., 1995) contains features extracted from breast mass images, which are then used to classify tumors as malignant or benign. See here for more information about the dataset. We fit each model on train sizes ranging from 25 to 400 examples, and tested on a held-out size of 169.

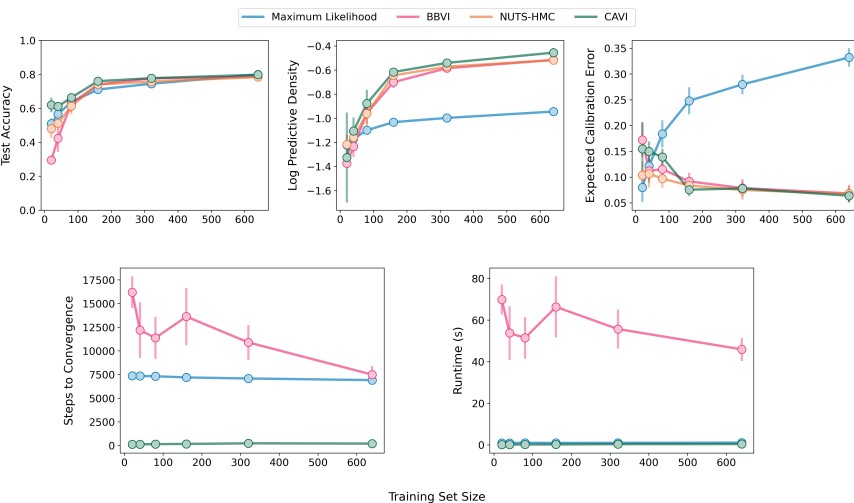

Figure 5: Performance and runtime results of the different models on the 'Vehicle Silhouettes' dataset. Descriptions of each subplot are same as in the Figure 2 legend.

### D.6    SONAR (MINES VS ROCKS) DATASET

The Sonar (Mines vs Rocks) dataset consists of sonar signals bounced off metal cylinders and rocks under various conditions. The dataset includes 111 patterns from metal cylinders (mines) and 97 patterns from rocks. Each pattern is represented by 60 continuous attributes corresponding to the energy within specific frequency bands (Sejnowski & Gorman). The task is to classify each pattern as either a mine (M) or a rock (R). For this dataset, we fit each model on train sizes ranging from 8 to 128 examples and tested on a held-out size of 80 examples. See here for more information about the dataset.

### D.7    BANKNOTE AUTHENTICATION DATASET

The 'Banknote Authentication' dataset (Lohweg, 2013) contains features extracted from images of genuine and forged banknotes. It is primarily used for binary classification tasks to distinguish between authentic and counterfeit banknotes. See here for more information about the dataset.

## E    UCI PERFORMANCE RESULTS

In Figures 5 to 9 we report the same performance and runtime metrics as in Figure 2 for 7 UCI datasets, and find that with the exception of the Sonar dataset, CAVI performs competitively with or better than MLE on all datasets, and always outperforms MLE in terms of LPD and ECE. Runtime scaling is similar as reported for the Pinwheels dataset in the main text; CAVI-CMN always converges in fewer steps and is faster than BBVI, and either outperforms or is competitive with MLE in terms of runtime.

## F    MODEL CONVERGENCE DETERMINATION

For each inference algorithm, the number of iterations taken to converge was determined by running each algorithm for a sufficiently high number of gradient (respectively, CAVI update) steps such that the ELBO (or log likelihood - LL - for MLE) stopped significantly changing. This was determined (through anecdotal inspection over many different initializations and runs across the different UCI datasets) to be 20,000 gradient steps for BBVI, 20,000 gradient steps for MLE, and 500 combined CAVI update steps for CAVI-CMN. To determine the time taken to sufficiently converge, we recorded the value of the ELBO or LL at each iteration, and fit an exponential decay function to

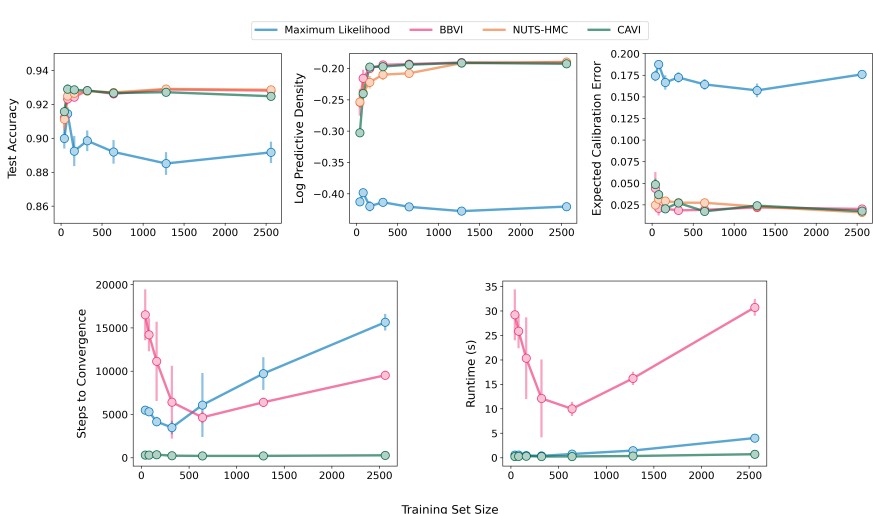

Figure 6: Performance and runtime results of the different models on the 'Rice' dataset. Descriptions of each subplot are same as in the Figure 2 legend.

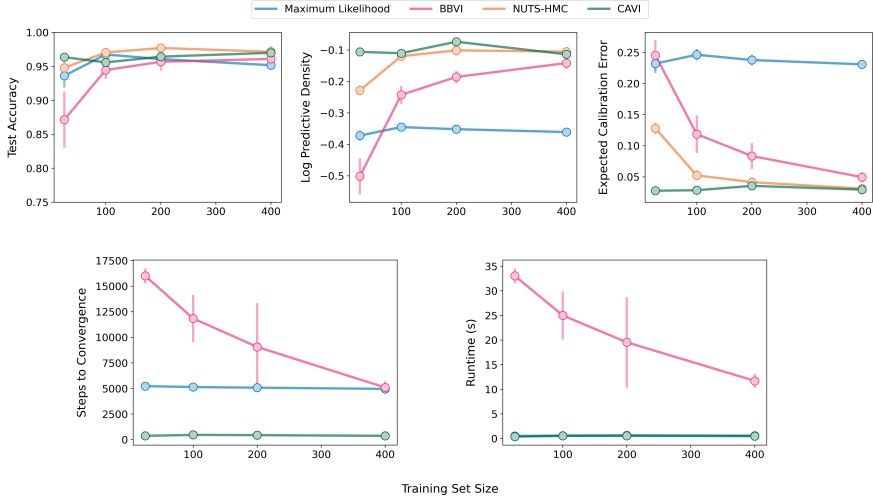

Figure 7: Performance and runtime results of the different models on the 'Breast Cancer' dataset. Descriptions of each subplot are same as in the Figure 2 legend.

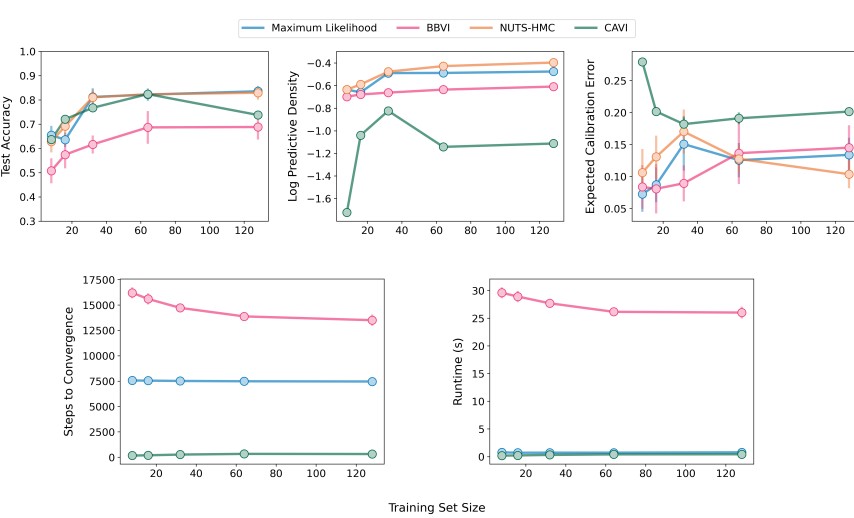

Figure 8: Performance and runtime results of the different models on the 'Connectionist Bench (Sonar, Mines vs. Rocks)' dataset. Descriptions of each subplot are same as in the Figure 2 legend..

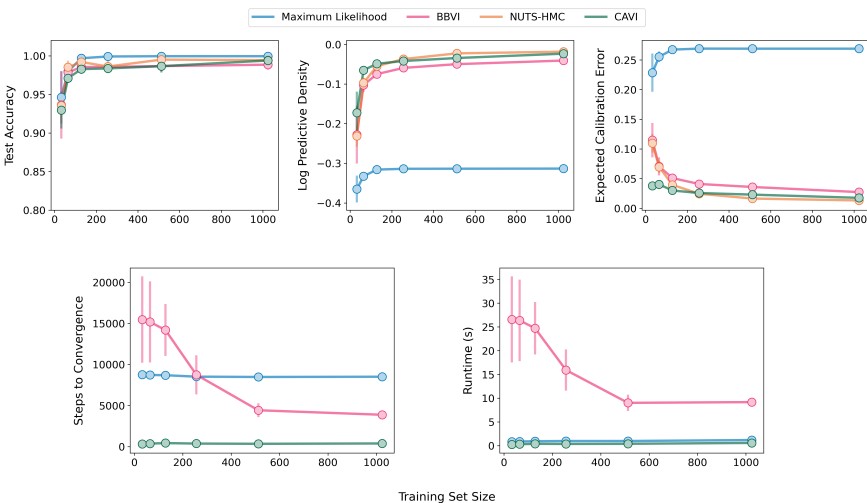

Figure 9: Performance and runtime results of the different models on the 'Banknote Authentication' dataset. Descriptions of each subplot are same as in the Figure 2 legend.

the negative of each curve. The parameters of the estimated exponential decay were then used to determine the time at which the curve decayed to 95% decay of its value. This time was reported as the number of steps taken to converge.

