# OpenReview forum: "Gradient-free variational learning with conditional mixture networks"
_ICLR.cc/2025/Conference — ICLR 2025 Conference Withdrawn Submission_

### Official Review · Reviewer_Ckkr · 2024-10-20

**Soundness:** 1
**Presentation:** 2
**Contribution:** 3
**Rating:** 3
**Confidence:** 5

**Summary:**

This paper proposes a model called the “conditional mixture network” (CMNs).  The idea is that a mixture of linear models is defined such that a given feature vector is transformed by one of the components.  This creates a hidden representation ($X_1$) that is then used as the features for a final linear model, which makes the classification decision.  This model deviates from predecessors such as mixture density networks and mixtures of experts as they usually define the mixture on the output distribution over the response / label.  CMNs, on the other hand, define the mixture to be on the (linear) transformation between the input ($X_0$) and first hidden layers ($X_1$).  The paper claims that the major benefit of this model over traditional Bayesian neural networks is that parameter estimation can be done without gradient-based stochastic optimization.  Rather, as the two-layer CMN admits conditional conjugacy, a mean-field coordinate ascent algorithm can be devised to estimate the posterior distribution (Section 3.4).  The experiments focus on validating that this coordinate ascent algorithm is superior to maximum likelihood estimation (MLE), black-box variational inference (BBVI), and Hamiltonian Monte Carlo (as implemented by NUTS).  Results are reported on synthetic data (Pinwheels and Waveform Domains) and on seven datasets from the UCI repository (Rice, Breast Cancer, Waveform, Vehicle Silh, Banknote, Sonar, and Iris).  In general, the coordinate ascent algorithm is reported to match the predictive performance of BBVI and NUTS while, like BBVI and NUTS, being superior to MLE.  Moreover, the coordinate ascent VI algorithm has a far superior runtime to NUTS and faster convergence compared to BBVI.

**Strengths:**

**Generative model of piecewise linear representations**:  Forming representations from piecewise linear functions is a core component of modern deep learning architectures—with the best example being ReLU activations.  This work presents an interesting generative perspective of these representations.  While not explored in this paper, I could see this leading to a better way to specify informative priors for feature learning (i.e. draw the linear models from distributions over particular basis function or groups).

**Weaknesses:**

1.  **Conflict between method’s presentation and experimental evaluation**:  The Introduction and beginning of Section 3.1 clearly motivates the work from the perspective of improving the uncertainty quantification abilities of deep learning architectures.  Thus, when arriving at the experiments section (4), I was expecting the result to compare the performance of the CMN with natural baselines, such as a two-layer NN with ReLU activations, since this would be a non-Bayesian piecewise linear model, and a two-layer Bayesian NN.  However, the experimental section’s main hypothesis is to examine if the coordinate ascent VI algorithm improves upon NUTS and BBVI.  While this is certainly a sensible and necessary experimental comparison to make, I don’t see how this line of experimentation alone is sufficient.  As far as I understand (see corresponding question #1 below), this is the first paper to describe the CMN model.  Hence, the generative model itself needs to be justified as well as any particular choice of inference algorithm.  I believe that adding comparisons to small Bayesian and non-Bayesian neural architectures would go a long way to justifying the CMN.


2.  **Black-box variational inference is not representative of the state of the art**:  In line 66, the paper claims that the coordinate ascent algorithm is benchmarked against “state-of-the-art Bayesian methods like NUTS and BBVI”.  I agree that NUTS is representative of the SOTA for Hamiltonian Monte Carlo implementations.  However, BBVI is not, as using REINFORCE to estimate the gradients usually has much higher variance than more modern alternatives.  The paper should also compare with differentiable non-centered parameterizations for the weights [Kingma & Welling, ICML 2014] (a.k.a. “Bayes by Backdrop” [Blundell et al., ICML 2015]).  The mixture gating variable will need additional tricks for differentiating through the discrete variable [Graves, 2016; Maddison et al., 2017].  There are also non-stochastic alternatives [Wu et al., ICLR 2019].  For a method that blends the benefits of variational inference and MCMC, I would also look at particle methods such as Stein Variational Gradient Descent [Liu & Wang, 2016].


3.  **Weak justification for why a mixture is a good choice for representation learning**:  As stated above, the related works section draws parallels to mixture density networks and mixtures of experts.  However, these models define mixtures over the output distribution, and I don’t think the same intuition can be extended to motivate why mixtures of linear models is the best construction for the hidden representation.  For example, using $Z_1,d$ to denote the gating variable for the $d$th hidden dimension, one could also place the prior $Z_{1,d} \sim \text{Bernoulli}(p)$.  This would have the same representational capacity as the propose CMN (linear transformation) but allow model capacity to be directly controlled by $p$.  This construction would also be similar to ReLU NNs since one can think of a subset of the linear models activating to define the transformation.  There might also be some interesting connection to dropout regularization.  I’m not saying that this dimension-wise Bernoulli construction is obviously better; I’m just saying that the paper should address why the mixture is better motivated than some obvious alternatives.  This relates back to weakness #1.


4.  **No discussion of extending the architecture to increased depths**:  While the paper argues that CMNs could serve as a building block for deeper architectures (see paragraph before the start of 3.2), there is no discussion of how this might be done from the generative perspective or for the inference algorithms.


Blundell, Charles, Julien Cornebise, Koray Kavukcuoglu, and Daan Wierstra. "Weight uncertainty in neural networks." In Proceedings of the 32nd International Conference on International Conference on Machine Learning-Volume 37, pp. 1613-1622. 2015.

Graves, Alex. "Stochastic backpropagation through mixture density distributions." arXiv preprint arXiv:1607.05690 (2016).

Kingma, Diederik, and Max Welling. "Efficient gradient-based inference through transformations between bayes nets and neural nets." International Conference on Machine Learning. PMLR, 2014

Liu, Qiang, and Dilin Wang. "Stein variational gradient descent: A general purpose bayesian inference algorithm." Advances in neural information processing systems 29 (2016).

Maddison, Chris J., Andriy Mnih, and Yee Whye Teh. "The Concrete Distribution: A Continuous Relaxation of Discrete Random Variables." International Conference on Learning Representations. 2017

Wu, Anqi, Sebastian Nowozin, Edward Meeds, Richard E. Turner, José Miguel Hernández-Lobato, and Alexander L. Gaunt. "Deterministic Variational Inference for Robust Bayesian Neural Networks." In International Conference on Learning Representations. 2019.

**Questions:**

1.  Is this the first paper to propose the conditional mixture network?  If not, please give the citation.  If so, please explain why no experimental comparisons to Bayesian and non-Bayesian ReLU NNs are given.


2.  Normalization layers (BatchNorm, LayerNorm) are often necessary for modern deep learning architectures.  Is there a way to incorporate them within the CMN?  And even better yet, define them within the same generative framework?


3.  Why is gradient-based learning spoken of so critically in the paper?  In the past, it was largely believed that coordinate ascent algorithms would be better than noisy, gradient-based ones, but with our modern knowledge about how many critical points are not minima and the benefits of noise during optimization, I don’t think many of those past critiques still hold today.

---

### Official Review · Reviewer_r9cp · 2024-10-23

**Soundness:** 2
**Presentation:** 1
**Contribution:** 1
**Rating:** 3
**Confidence:** 3

**Summary:**

The paper presents a method called CAVI-CMN, a gradient-free learning algorithm using conditional mixture networks (CMNs) for Bayesian inference.
The method applies coordinate ascent variational inference (CAVI) to a probabilistic mixture-of-experts model, aiming to balance expressiveness, scalability, and uncertainty quantification in classification tasks.
The authors propose the use of conjugate priors and Polya-Gamma augmentation to allow for efficient inference without gradient-based optimization.
Experiments on synthetic and small UCI datasets demonstrate some competitiveness compared to alternatives like maximum likelihood estimation (MLE) and black-box variational inference (BBVI).

**Strengths:**

The paper introduces a method that eliminates the need for gradient-based optimization, which is a significant advantage for reducing computational costs.

**Weaknesses:**

- Heavy notation: The notation is often overly complex, and impacts readability and understanding. It lacks consistency across sections, making it difficult to follow key points. For instance, the sadly non-numbered equation on page 3 is very unclear:

     1. on line 150, authors define $p_k$, yet this does not appear (at least explicitly) in the equation above

     2. there is no explanation nor definition about $\pi$. What is $p(\pi)$? and $p(z^n|\pi)$?

     3. on line 150, authors say that y^n is an observation, yet on line 49 and 168, y is intended to be a label.

     4. x does not appear in the LHS of the equation, nor in the explanation below it, but only on the RHS.

Overall, this confusion makes the problem setting very unclear, and impacts next derivations.

- The goal of the paper is somewhat puzzling. It is unclear whether the primary focus is computational efficiency, uncertainty quantification, or scalability, as the paper lacks a clear articulation of its main contributions.

- Small datasets: The experiments are performed on small datasets from the UCI repository, spanning from 4 up to only 60 features.

- I'm not an expert in this field, but I have the feeling that relevant literature is missing, as most of the references in the paper are quite old. In particular, the two main competitors for the experiments, NUTS and BBVI, both date back to 2014.


minors:
- Table 1 does not report somehow the best results, and it is hard to get a message from it.
- it is unclear to me how the first layer of the network is able to output a joint continuous-discrete output (L177).

**Questions:**

- Why were the experiments limited to such small datasets? Would CAVI-CMN still be competitive in terms of performance and runtime on larger, modern datasets?

- How does the method compare with more recent advancements in variational inference or Bayesian deep learning, particularly those leveraging deep architectures and large-scale datasets?

- Could the authors elaborate on the practical use cases where CAVI-CMN would be preferred over simpler, gradient-based methods, given its specialized nature?

---

### Official Review · Reviewer_XKV8 · 2024-11-03

**Soundness:** 3
**Presentation:** 3
**Contribution:** 2
**Rating:** 5
**Confidence:** 4

**Summary:**

The submission introduces a coordinate ascent variational inference (CAVI) approach for learning a mixture of linear transformations of an input vector to predict a label, termed conditional mixture network (CMN). A Poly-Gamma augmentation scheme is considered to arrive at a conditionally conjugate model as commonly required for CAVI. The suggested approach appears competitive to black-box VI or NUTS in terms of model calibration, while requiring less compute time.

**Strengths:**

While Poly-Gamma augmentations have been used for models with other likelihoods or for Gibbs samplers, their usage for multinomial logistic regression models is new as far as I am aware. The paper aims to address an important problem in that discriminative classifiers can yield not well-calibrated predictions when trained to maximize the log-likelihood, while Bayesian approaches can be computationally expensive. In their arguably simple examples, the authors demonstrate that their approach does indeed improve BBVI or NUTS in terms of calibration or WAIC.

**Weaknesses:**

The empirical evaluation seems a bit limited. In particular, it would useful if the authors would consider additional baselines. In particular \
(i) variational approaches beyond BBVI for the same model, or at least with some standard control variates [1,2,3], \
(ii) stochastic-gradient MCMC algorithms [4,5] that can yield better uncertainties [6,7] and are more scalable than NUTS, \
(iii) different models (e.g. deep ensembles [8], BNNs) to see better assess how the reported results (accuracies, expected calibration error, etc) compare to previous models.

The authors claim that ‘CAVI-CMT achieves SOTA-like performance with absolute runtime comparable to a backpropagation-based MLE approach’. However, the used MLE baseline does not use stochastic gradients (i.e. mini-batches), which is arguably a more common approach, and I expect to converge faster?

The authors claim that ‘using fully conjugate priors within the CAVI framework does not diminish the inference and predictive performance’. However, it is not clear to me why different choices of priors would perform. Does the BBVI or MCMC baseline use different priors?



[1] Tucker, George, et al. "Rebar: Low-variance, unbiased gradient estimates for discrete latent variable models." Advances in Neural Information Processing Systems 30 (2017).\
[2] Grathwohl, Will, et al. "Backpropagation through the Void: Optimizing control variates for black-box gradient estimation." International Conference on Learning Representations. 2018.\
[3] Richter, Lorenz, et al. "VarGrad: a low-variance gradient estimator for variational inference." Advances in Neural Information Processing Systems 33 (2020): 13481-13492.\
[4] Welling, Max, and Yee W. Teh. "Bayesian learning via stochastic gradient Langevin dynamics." Proceedings of the 28th international conference on machine learning (ICML-11). 2011.\
[5] Chen, Tianqi, Emily Fox, and Carlos Guestrin. "Stochastic gradient hamiltonian monte carlo." International conference on machine learning. PMLR, 2014.\
[6] Zhang, Ruqi, et al. "Cyclical Stochastic Gradient MCMC for Bayesian Deep Learning." International Conference on Learning Representations, 2019.\
[7] Maddox, Wesley J., et al. "A simple baseline for bayesian uncertainty in deep learning." Advances in neural information processing systems 32 (2019).\
[8] Lakshminarayanan, Balaji, Alexander Pritzel, and Charles Blundell. "Simple and scalable predictive uncertainty estimation using deep ensembles." Advances in neural information processing systems 30 (2017).

**Questions:**

It is not clear to me whether the lower bounds for CAVI higher than the BBVI-ELBO in the experiments? If not, why does it yield better results in terms of calibration etc? Is this due to the choice of priors, or due to the biases of the variational distribution with the Poly-Gamma augmentation?

Why is there a Gaussian prior for the bias term u in line 181, which seems to be absent in the subsequent model description?

---

### Official Review · Reviewer_u8kz · 2024-11-03

**Soundness:** 3
**Presentation:** 2
**Contribution:** 3
**Rating:** 5
**Confidence:** 4

**Summary:**

The paper introduces Coordinate Ascent Variational Inference for Conditional Mixture Networks (CAVI-CMNs), a framework designed to achieve efficient, scalable Bayesian inference for probabilistic models, specifically focusing on CMNs. By leveraging conditional conjugacy and Pólya-Gamma augmentation, this approach enables effective variational updates without relying on gradient-based optimization, positioning it as a computationally viable alternative to traditional Bayesian inference methods such as Maximum Likelihood Estimation, Black-Box Variational Inference, and the No-U-Turn Sampler.  The model's performance is validated on two synthetic datasets (Pinwheels and Waveform Domains) and six real-world classification datasets from the UCI Machine Learning Repository, demonstrating that it maintains competitive predictive accuracy and runtime efficiency even with increasing model complexity. This efficiency is particularly advantageous for applications requiring rapid, uncertainty-aware predictions, including in fields such as autonomous systems, healthcare, and finance.

**Strengths:**

- The CAVI-CMNs approach introduces a highly efficient gradient-free Bayesian inference method specifically designed for CMNs. Its reliance CAVI paired with conjugate priors and Pólya-Gamma augmentation, demonstrates innovation through a novel combination of existing ideas, especially for probabilistic models that typically depend on gradient-based methods. This efficiency could mark an important step forward in Bayesian neural network inference.

- The paper ensures reproducibility by including comprehensive instructions and corresponding Python codes for nearly all numerical experiments.

-  The paper proposes future directions for extending CAVI-CMNs to support deeper architectures and online learning contexts, demonstrating foresight in adapting the method for broader applicability. Such adaptability may foster additional research and application of CAVI-CMNs, underscoring its potential for significant impact.

**Weaknesses:**

**Insufficient Experimental Validation:** The reliance on smaller UCI and synthetic datasets might not adequately demonstrate the robustness or scalability of the approach in real-world, high-dimensional applications. Without validation on challenging or large-scale datasets, such as the CIFAR-10 dataset[1] and the ImageNet[2] or real-time data streams, the method’s claims of scalability and efficiency could be viewed as unconvincing, limiting the impact and generalizability of the results.

**Enhance Consistency and Clarity in Mathematical Notation:**

+ The sample size should be consistently denoted as $N$ throughout the paper. For example, Line 049 uses $n$, whereas Line 147 switches to $N$.

+ Lines 147-153 contain an indexing typo where the index should be written as $n=1$ instead of $i=1$.

+ Line 168: The notation for multivariate input and scalar output is inconsistent, with input variables in bold and outputs in regular font. Adopting a consistent notation style, as outlined in the ICLR guidelines, would improve clarity.

+ **Line 246-256**: Could the authors clarify the specific distributions, such as $\mathcal{MN}$ and $\Gamma$? It would enhance readability if the full names or definitions of these distributions were provided upon first mention. Additionally, would a more flexible prior for $\mathbf{\Sigma}_k$ improve the model's performance?

+ **Line 246-256**: $p(Y|\mathbf{X})$ should be $p(Y,\mathbf{X})?$

- **Baseline Comparison**: **Line 322-323**: For parameter point estimates, in addition to comparing with Maximum Likelihood Estimation using backpropagation, the authors should consider comparing their AVCI (variational EM algorithm) with a standard EM algorithm. This comparison would help elucidate the advantages and limitations of the proposed method relative to traditional approaches, providing a clearer understanding of its effectiveness.

- **Lines 114 and 669-671**: The citations are incorrect. The correct citation should be: Bishop, Christopher M., and Markus Svenskn. "Bayesian Hierarchical Mixtures of Experts." UAI'03 Proceedings of the Nineteenth conference on Uncertainty in Artificial Intelligence. Morgan Kaufmann Publishers Inc., 2003.


References:

[1] Krizhevsky, A., \& Hinton, G. (2009). Learning multiple layers of features from tiny images.

[2] Deng, J., Dong, W., Socher, R., Li, L. J., Li, K., \& Fei-Fei, L. (2009, June). Imagenet: A large-scale hierarchical image database. In 2009 IEEE conference on computer vision and pattern recognition (pp. 248-255). Ieee.

**Questions:**

- **Broader Range of Gradient-Based Optimization Techniques**: To strengthen the comparative analysis, it would be beneficial to extend the comparison to include both [1] and the AdaBelief optimizer [2]. This broader range of gradient-based optimization techniques could provide deeper insights into the performance and robustness of the proposed method, offering a more comprehensive evaluation than focusing solely on AdaBelief [2].

- **Weak Theoretical Justification:** While CAVI-CMN effectively employs conditional conjugacy and Pólya-Gamma augmentation, the paper lacks theoretical justification or clear proofs regarding convergence rates and optimality, as seen in related works [5,6,7]. This absence may raise concerns about the reliability and consistency of CAVI-CMNs, particularly in complex models. Including such theoretical justification, alongside the presented empirical evidence, would substantially enhance the paper's contribution and support its novel approach.

- **Overly Simplistic Conditional Mixture of Experts**: Lines 179-183 point out that the current architecture is relatively simple, utilizing only a linear expert with discrete output and a pair of latent variables. It remains unclear how the proposed method would scale to more complex models, such as those with nonlinear experts, continuous outputs, or deeper architectural layers such as [6,7]. Can this approach be adapted to effectively handle these additional complexities?

- **Model Selection Architecture**: The paper lacks clarity on the criteria used for model selection, specifically regarding the number of experts $K$ and the dimensionality of the latent space $h$ for real datasets. Providing a detailed explanation or methodology for selecting these parameters would enhance understanding of the architecture’s adaptability and effectiveness, especially in practical applications.

- **Scalability Challenges and Potential Extensions for CAVI-CMN in High-Dimensional Architectures:** Figure 4 suggests that as the dimensionality of the latent variable $X_1$ increases, CAVI-CMN exhibits more pronounced scaling challenges than the other two methods as the latent dimension $h$ grows. This raises questions about the current method's adaptability to deeper (additional layers) and wider (larger latent dimensions) architectures. The authors should provide additional details on how incorporating techniques like low-rank approximations or further factorization could enhance the model’s scalability. Further investigation and simulations demonstrating the impact of these approaches would offer valuable insights into the model's ability to handle increasingly complex, high-dimensional architectures efficiently.

References:

[1] Xie, X., Zhou, P., Li, H., Lin, Z., \& Yan, S. (2024). Adan: Adaptive nesterov momentum algorithm for faster optimizing deep models. IEEE Transactions on Pattern Analysis and Machine Intelligence.

[2] Zhuang, J., Tang, T., Ding, Y., Tatikonda, S. C., Dvornek, N., Papademetris, X., \& Duncan, J. (2020). Adabelief optimizer: Adapting stepsizes by the belief in observed gradients. Advances in neural information processing systems, 33, 18795-18806.

[3] Bhattacharya, A., Pati, D., \& Yang, Y. (2023). On the convergence of coordinate ascent variational inference. arXiv preprint arXiv:2306.01122.

[4] Zhang, F., \& Gao, C. (2020). Convergence rates of variational posterior distributions. The Annals of Statistics, 48(4), 2180-2207.

[5] Wang, B., \& Titterington, D. M. (2006). Convergence properties of a general algorithm for calculating variational Bayesian estimates for a normal mixture model. Bayesian Analysis, 1(3), 625-650.

[6] Viroli, C., \& McLachlan, G. J. (2019). Deep Gaussian mixture models. Statistics and Computing, 29, 43-51.

[7] Kock, L., Klein, N., \& Nott, D. J. (2022). Variational inference and sparsity in high-dimensional deep Gaussian mixture models. Statistics and Computing, 32(5), 70.

---

### Note · Authors · 2024-11-28

I have read and agree with the venue's withdrawal policy on behalf of myself and my co-authors.